# A deep learning model to predict RNA-Seq expression of tumours from whole slide images

Benoît Schmauch [1✉], Alberto Romagnoni[1,4], Elodie Pronier[1,4], Charlie Saillard[1], Pascale Maillé[2,3], Julien Calderaro[2,3], Aurélie Kamoun [1], Meriem Sefta[1], Sylvain Toldo[1], Mikhail Zaslavskiy[1], Thomas Clozel [1], Matahi Moarii[1], Pierre Courtiol[1,5] & Gilles Wainrib[1,5✉]

Deep learning methods for digital pathology analysis are an effective way to address multiple clinical questions, from diagnosis to prediction of treatment outcomes. These methods have also been used to predict gene mutations from pathology images, but no comprehensive evaluation of their potential for extracting molecular features from histology slides has yet been performed. We show that HE2RNA, a model based on the integration of multiple data modes, can be trained to systematically predict RNA-Seq profiles from whole-slide images alone, without expert annotation. Through its interpretable design, HE2RNA provides virtual spatialization of gene expression, as validated by CD3- and CD20-staining on an independent dataset. The transcriptomic representation learned by HE2RNA can also be transferred on other datasets, even of small size, to increase prediction performance for specific molecular phenotypes. We illustrate the use of this approach in clinical diagnosis purposes such as the identification of tumors with microsatellite instability.

[1] Owkin Lab, Owkin, Inc., New York, NY, USA. [2] INSERM U955, Team "Pathophysiology and Therapy of Chronic Viral Hepatitis and Related Cancers", Créteil, France. [3] APHP, Department of Pathology, Hôpital Henri Mondor, Université Paris-Est, Créteil, France. [4]These authors contributed equally: Alberto Romagnoni, Elodie Pronier. [5]These authors jointly supervised this work: Pierre Courtiol, Gilles Wainrib. ✉email: benoit.schmauch@owkin.com; gilles.wainrib@owkin.com

Histological analyses of tumor biopsy sections are important tools in oncology, providing a high-resolution map of the tumor that helps pathologists determining diagnosis and grade[1,2]. Technological progress and growing availability of large datasets have made it possible to train increasingly sophisticated algorithms, which can process and learn from high-definition whole-slide images (WSI). Convolutional neural networks (CNNs) have recently emerged as an important image analysis tool, accelerating the work of pathologists. They have shattered performance in many challenging clinical diagnosis applications, including mitosis detection[3], quantification of tumor immune infiltration[4], cancer subtypes classification[5], and grading[6]. These tools emerge as exciting opportunities in the clinical and biomedical field[7], ultimately improving the prediction of patient survival outcomes and response to treatment[8,9].

While it is becoming clear that the application of such models to tissue-based pathology can be very useful, few attempts have been made to connect specific molecular signatures directly to morphological patterns. Recent studies have shown how models of this class can connect histological images to tumor-specific mutations or tumor mutational burden in the lung[8], prostate[10], brain cancers[11], and melanoma[12,13]. Massive changes in gene expression are known to occur in many cancers secondary to mutations or epigenomic modifications, and the comprehensive characterization of disease-related gene signatures can help to clarify disease mechanisms and prioritize targets for novel therapeutic approaches[14,15]. Whole-transcriptome sequencing techniques (RNA-Seq) have been developed, together with dedicated bioinformatic tools[16,17], to reconstruct gene information in carcinogenesis[18,19]. Despite their sustained decrease in cost[20], these technologies are not routinely used by all medical centers. They are time-consuming and present several challenges impairing their adoption into clinical practice. Gene expression is highly variable and particularly affected by cell type, proliferation, and differentiation status[21,22]. However, the possibility of quantifying the expression levels of specific genes based on a visual observation of hematoxylin & eosin (H&E)-stained WSI has never been investigated in detail. Predicting gene expression from WSIs would greatly facilitate patient diagnosis and prediction of response to treatment and survival outcome[23].

Here, we present HE2RNA, a deep-learning algorithm specifically customized for the prediction of gene expression from WSI (Fig. 1). For training our model, we collected WSIs and their corresponding RNA-Seq data from The Cancer Genome Atlas (TCGA) public database. We then investigated how HE2RNA could be used to generate heatmaps for a spatial visualization of any gene expression. Finally, we show how the internal representation (transcriptomic representation) learned by the model can improve the prediction of a specific molecular phenotype such as microsatellite instability.

## Results

**A deep-learning model for the prediction of gene expression.** We used matched WSIs and RNA-Seq profiles from TCGA data (https://portal.gdc.cancer.gov/), including 8725 patients and 28 different cancer types, to develop HE2RNA, a deep-learning model based on a multitask weakly supervised approach[24] (architecture in the "Methods"). The model was trained to predict normalized gene expression data (logarithmic FPKM-UQ values, see "Methods") from WSIs. We performed a five-fold cross-validation, i.e., patients were randomly assigned to five different sets, and each set was used in turn as the validation set, the other four sets being used for training. The final results were expressed as a mean over all five runs (see "Methods").

We restricted our analysis to 30,839 (coding/noncoding) genes with nonzero median expression (see "Methods"). WSIs were partitioned into "tiles" (squares of $112 \times 112 \ \mu m$) and aggregated into clusters, called supertiles. The number and size of these supertiles were optimized for each specific task detailed thereafter. For the transcriptome prediction task, 100 supertiles were created for each WSI. A multilayer perceptron was applied to all supertiles to generate a predicted value per gene and per supertile. For comparison of the model predictions with the real RNA-Seq value, the predictions per super-tile were aggregated by calculating a weighted mean to give a final prediction per-WSI (see "Methods" and Supplementary Fig. 1).

This correlation was assessed for each gene, separately for each different type of cancer. We considered a prediction to be significantly different from the random baseline value if the $p$-value associated with its coefficient $R$ was below 0.05, after applying Holm–Šidák (HS) or Benjamini–Hochberg (BH) correction to account for multiple-hypothesis testing. An average of 3627 genes (respectively 12,853), including 2797 protein-coding (respectively 8450) per cancer type were predicted with a statistically significant correlation under HS correction (Fig. 2) (respectively under BH adjustment, Supplementary Fig. 2).

The number of significantly well-predicted genes varied considerably between cancer types, mostly due to the size of the dataset considered (Fig. 2a): the smaller the number of samples, the higher the correlation coefficient required for statistical significance. For example, under HS correction, only seven genes were accurately predicted for the 44 samples of diffuse large B-cell lymphoma (DLBC), ($R > R_{sign} = 0.64$), whereas 15,391 genes were correctly predicted for the 1046 samples of lung carcinoma (indicated as LUNG, including 535 WSIs for lung adenocarcinoma—LUAD—and 511 slides for lung squamous cell carcinoma—LUSC), ($R > R_{sign} = 0.20$).

We compared the list of genes well-predicted in each cancer to analyze the consistency of the predictions. None of the genes were well-predicted in all 28 available cancer types (Fig. 2b), but few genes were consistently above the significance threshold when considering smaller subsets of cancer. In particular, *C1QB* expression was strikingly well-predicted in 17/28 different cancer datasets ($R = 0.39 \pm 0.15$). Similarly, *NKG7*, *ARHGAP9*, *C1QA*, and *CD53* were accurately predicted in 15/28 datasets ($R = 0.38$–$0.46$ for the various cancer types). C1QA and C1QB are proteins of the complement known to be involved in T-cell activation following antigen presentation by antigen-presenting cells (APC)[25], whereas CD53[26] and NKG7[27] are known to be expressed by T and NK cells, respectively.

Longer lists of genes were consistently well-predicted by HE2RNA in smaller subsets of cancer types, and we used ingenuity pathway analysis (IPA) software to identify the corresponding biological networks. We found 156 genes that were well-predicted separately in at least 12 out of 28 different cancer types. For this subset of genes, we performed a functional annotation (Fig. 2c).

This analysis revealed an enrichment in genes involved in immunity and T-cell regulation, as already suggested by the genes mentioned above. Indeed, the most significant functional network was the Th1–Th2 activation pathway ($p$-value $= 7.94 \times 10^{-15}$, right-tailed Fisher's exact test), iCOS-iCOSL signaling in T-helper cells, T-cell receptor signaling and CD28 signaling in T-helper cells. Given the large variability of gene expression profiles between different cancer types, we performed a similar analysis on two different cancers, namely liver hepatocellular carcinoma (LIHC) and invasive breast carcinoma (BRCA). In LIHC, the genes for which expression was most accurately predicted were associated with mitosis and cell-cycle control (cell-cycle control of chromosomal replication, mitotic roles of polo-like kinase), known hallmarks of cancer (Fig. 2d). Hepatic fibrosis, a known risk factor for the development of LIHC[28], was also among the

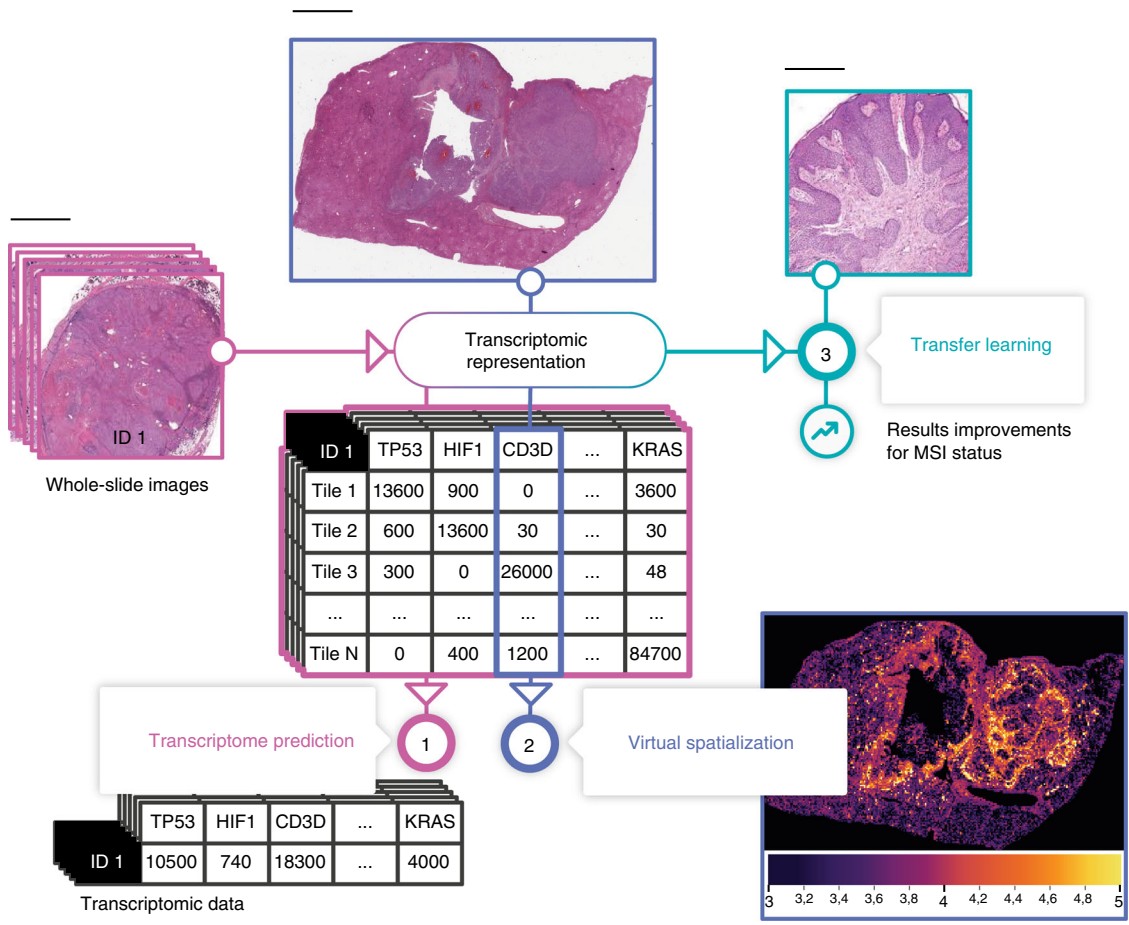

**Fig. 1 Graphical abstract: Transcriptomic learning for digital pathology.** Hematoxylin & eosin (H&E)-stained histology slides and RNA-Seq data (FPKM-UQ values) for 28 different cancer types and 8725 patients were collected from The Cancer Genome Atlas (TCGA) and used to train the neural network HE2RNA to predict transcriptomic profile from the corresponding high-definition whole-slide images (WSI). During this task, the neural network learned an internal representation encoding both information from tiled images and gene expression levels. This transcriptomic representation can be used for: (1) transcriptome prediction from images without associated RNA sequencing. (2) The virtual spatialization of transcriptomic data. For each predicted coding or noncoding gene, a score is calculated for each tile on the corresponding WSI, which can be interpreted as the predicted gene expression for this tile (even though the real value is available only for the slide). These predictive scores can be used to generate heatmaps for each gene for which expression is significantly predicted. (3) Improving predictive performances for different tasks, in a transfer learning framework, as shown here for a realistic setup, for microsatellite instability (MSI) status prediction from non-annotated WSIs. Scale bar: 5 mm.

most significantly well-predicted networks. Similarly, we found that the model performed well on BRCA samples (Fig. 2e) for the prediction of expression levels for genes involved in cell-cycle regulation (cell cycle: G2/M DNA damage checkpoint regulation, cell-cycle control of chromosomal replication, mitotic roles of polo-like Kinase), but also for the prediction of expression levels for *CHEK2* (known to be mutated in BRCA[29] and involved in its progression[30]) and *Cyclin E* (known to be overexpressed in BRCA[31]). These results demonstrate that HE2RNA, despite training on a diverse range of cancer types, was able not only to predict expression levels for genes involved in immune regulation but also to detect pathways deregulated in specific types of cancer.

Finally, we investigated whether known gene signatures dysregulated in a majority of cancer types could be accurately predicted by HE2RNA. We focused on the hallmarks of cancer, corresponding to the six biological capabilities acquired during the multistep development of human tumors[32]. Based on these hallmarks, we combined several lists of genes from Gene Set Enrichment Analysis (GSEA) software for each of these biological networks, to obtain six lists of genes involved in pathways known to be deregulated in several cancer types: increased angiogenesis,

increased hypoxia, deregulation of the DNA repair system, increased cell-cycle activity, immune response mediated by B cells, and adaptive immune response mediated by T cells (see "Methods" and Supplementary Table 7).

HE2RNA was able to significantly predict the activity of each of these pathways. (Fig. 3a). We found that, in 50% of cancer types for angiogenesis, and 54% for hypoxia, DNA repair, and cell-cycle pathways, signatures were significantly better predicted by HE2RNA than random lists of genes, with these proportions reaching 75 and 86% for B- and T-cell-mediated immunity, respectively (Fig. 3a). Similarly, when comparing the proportion of well-predicted genes, HE2RNA predictions were significantly better than for a random set of genes in 36% (angiogenesis), 29% (hypoxia), 25% (DNA repair), 39% (cell cycle), 36% (B-cell-mediated immunity), and 50% (T-cell-mediated immunity) of cancer types (Fig. 3b).

As expected, the proportion of genes accurately predicted for the six pathways was higher in the largest datasets (BRCA and LUNG) (Supplementary Fig. 3). Nevertheless, HE2RNA predicted expression profiles for a significant proportion of genes within these pathways in some of the smallest datasets: 18% of the cell-cycle pathway genes were accurately predicted in

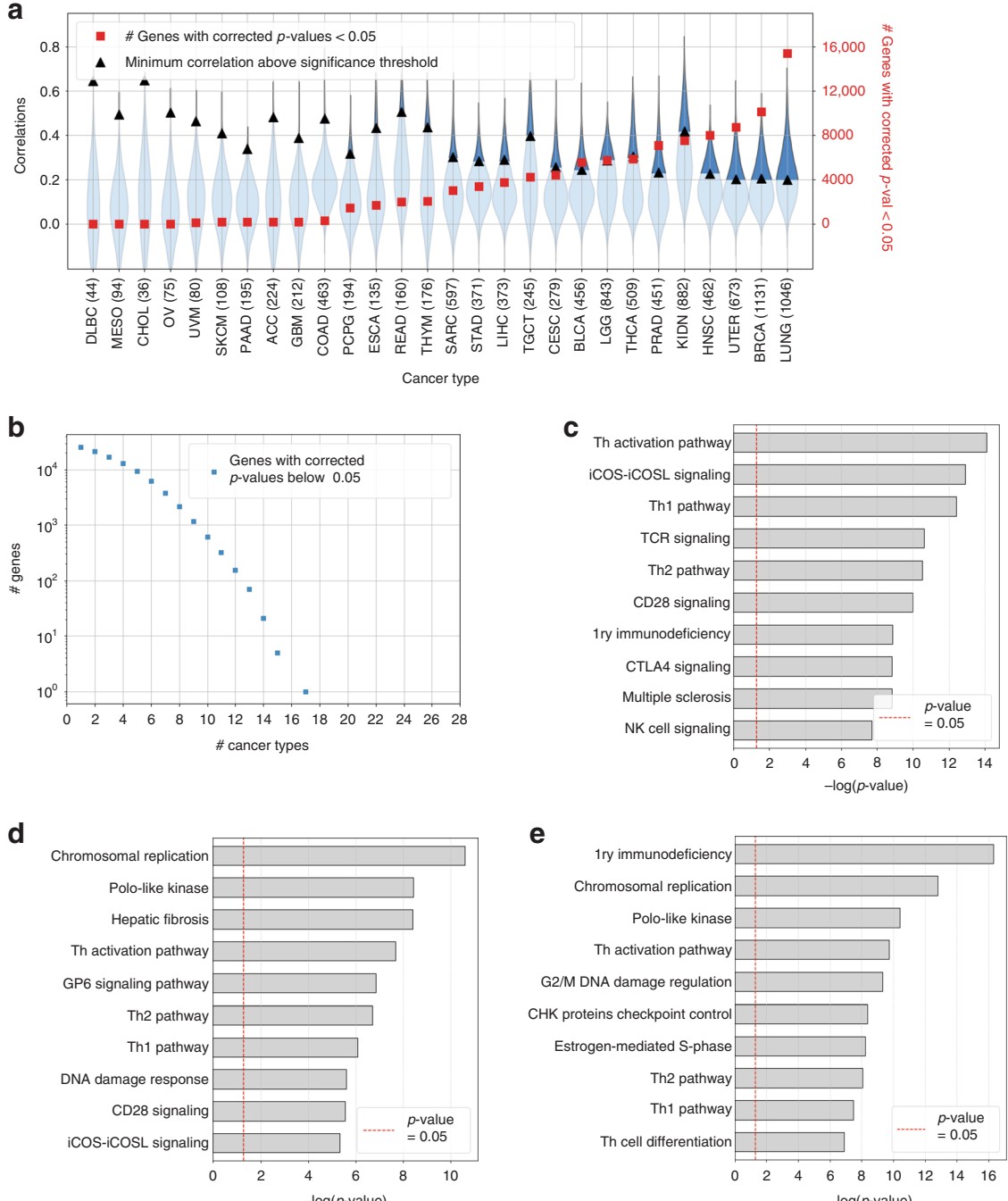

**Fig. 2 Gene expression prediction results. a** Distribution of Pearson correlation coefficients *R* averaged over the five folds of cross-validation (left axis, blue violin plots) and number of coding and noncoding genes (right axis, red squares) with Holm–Šidák corrected *p*-values < 0.05 (one-sided empirical *p*-value, as described in "Methods"), for 28 cancer types from the TCGA. Black triangles indicate the minimum correlation coefficient required for significance in any given dataset. **b** The number of coding and noncoding genes significantly well-predicted for a given number of cancer types, with Holm–Šidák corrected *p*-values < 0.05, as a function of the number of cancers. **c** Computational pathway analysis with ingenuity pathway analysis (IPA) software of the 156 best-predicted Pan TCGA genes, showing an enrichment in genes associated with immunity and tumor immune infiltration/activity. TCR: T-cell receptor, NK: natural killer. **d** IPA-based analysis of the more accurately predicted protein-coding genes in the LIHC dataset, showing an enrichment in genes associated with cell cycle and DNA damage response. LIHC: liver hepatocellular carcinoma. **e** IPA-based analysis of the more accurately predicted protein-coding genes in the BRCA dataset, showing an enrichment in genes associated with cell cycle and DNA damage response. Th cell differentiation: Th1 and Th2 cell differentiation. Th activation pathway: Th1 and Th2 activation pathway, 1ry: Primary. In **c**–**e** red dashed line = −log(*p*-values = 0.05), BRCA: breast cancer, *p*-values were calculated using right-tailed Fisher's exact test.

pancreatic adenocarcinoma (PAAD), 29% of the angiogenesis network, and 23% of the hypoxia pathway genes in pheochromocytoma and paraganglioma (PCPG). These results confirmed our previous analysis (Fig. 2e) in which genes involved

in cell-cycle regulation were among the most accurately predicted for the BRCA dataset.

As a control experiment, we used HE2RNA to predict the level of expression of housekeeping genes (HK) (Supplementary

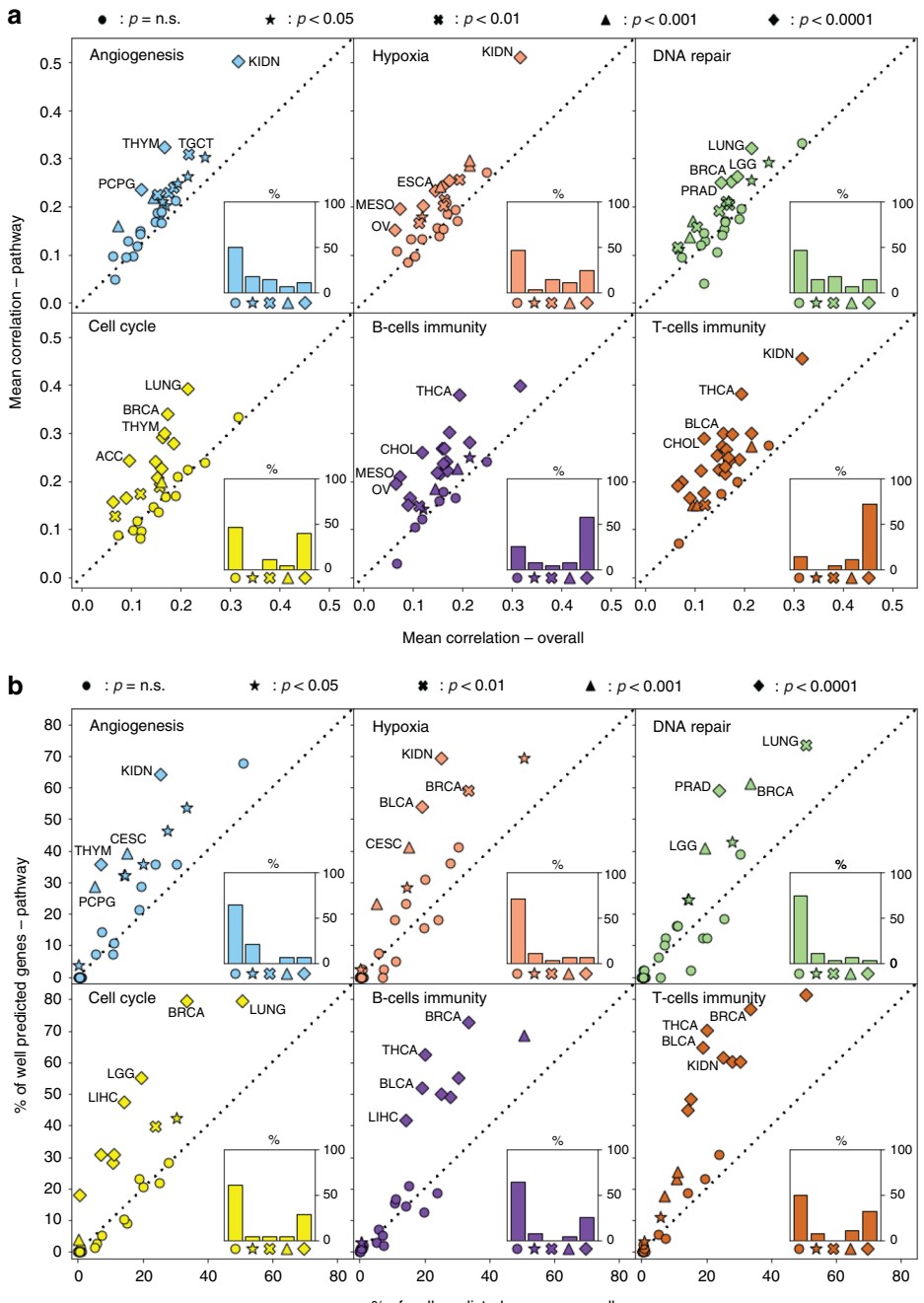

**Fig. 3 Prediction of signatures for cancer hallmarks. a** Comparison of correlation scores $R_p$ for each gene pathway defined in Supplementary Table 4 and involved in angiogenesis, hypoxia, DNA repair, cell cycle, and immune responses mediated by B and T cells, with the mean correlation coefficient $R_O$ obtained for 10,000 random lists of the same number of genes, for all 28 cancer types from the TCGA dataset. The indicated statistical significance refers to the probability of obtaining a correlation $R > R_p$ in the distribution of correlations for random lists, for each given cancer type. Insets show the percentages (%) of the different cases of statistical significance between cancer types. The dashed line is the identity line $R_p = R_O$. **b** As in **a**, but in terms of percentage of genes considered well-predicted (as defined in the text and in Fig. 2). One-sided empirical $p$-values computed as described in "Methods" (circle: $p = $ ns, star: $p < 0.05$, cross: $p < 0.01$, triangle: $p < 0.001$, square: $p < 0.0001$).

Table 7). We expected poor predictive performance for these genes as their expression levels have very little variations in diverse cell types under normal or pathological conditions. Indeed, the prediction performance for this set of genes was not significantly better than the performance for randomly selected genes (Supplementary Fig. 4). The correlation for the HK signature was significantly better than that for a random set of genes in only 14% of cancer types, and this percentage dropped to zero for the proportion of well-predicted genes. This analysis

validated our normalization procedure and the ability of HE2RNA to focus specifically on cancer-related molecular information.

**A tool for virtual spatialization**. HE2RNA assigns a score to all supertiles contributing to gene prediction and is therefore interpretable by design. Once a predictive model has been trained, it can identify the specific regions predictive of the expression level of a given gene on the WSIs. The larger the number of supertiles

**Table 1 Performance metrics for the prediction of genes of interest for virtual spatialization.**

| Cancer type | Gene | Correlation ($R$) | $p$-value (HS) | $p$-value (BH) |
|---|---|---|---|---|
| LIHC | CD3D | 0.40 | $<10^{-4}$ | $<10^{-4}$ |
| LIHC | CD3E | 0.41 | $<10^{-4}$ | $<10^{-4}$ |
| LIHC | CD3G | 0.41 | $<10^{-4}$ | $<10^{-4}$ |
| LIHC | CD247 | 0.37 | $<10^{-4}$ | $<10^{-4}$ |
| LIHC | CD19 | 0.32 | $<10^{-4}$ | $<10^{-4}$ |
| LIHC | CD20 | 0.27 | 0.58 | $2 \times 10^{-4}$ |
| LIHC | MKI67 | 0.47 | $<10^{-4}$ | $<10^{-4}$ |
| COAD | CD3D | 0.43 | 0.63 | $8 \times 10^{-4}$ |
| COAD | CD3E | 0.39 | 0.63 | $8 \times 10^{-4}$ |
| COAD | CD3G | 0.41 | 0.63 | $8 \times 10^{-4}$ |
| COAD | CD247 | 0.39 | 0.63 | $8 \times 10^{-4}$ |
| COAD | CD19 | 0.20 | ~1 | 0.009 |
| COAD | CD20 | 0.11 | ~1 | 0.07 |
| PRAD | TP63 | 0.18 | ~1 | 0.009 |
| PRAD | KRT8 | 0.12 | ~1 | 0.06 |
| PRAD | KRT18 | 0.12 | ~1 | 0.07 |

*LIHC* liver hepatocellular carcinoma, *COAD* colorectal carcinoma, *PRAD* prostate adenocarcinoma samples from TCGA dataset.
We used Pearson correlation, and one-sided empirical $p$-values corrected with Holm–Šidák (HS) and Benjamini–Hochberg (BH) correction for multiple-hypothesis testing.

chosen for the model, the higher the definition of the spatialization will be. The limit is reached when all the tiles of the training WSIs are treated separately ($k = 8000$) (Supplementary Fig. 1). Previous studies[31,33,34] have demonstrated that a virtual spatialization map (VSM), covering the entire WSI can be defined on the basis of CNN models. These heatmaps reflect the importance score assigned to each tile used in the algorithm.

We validated the accuracy of such VSMs on different datasets. First, we used a dataset containing tile images from 86 CRC slides, labeled with nine different classes: background, debris, adipose, smooth muscle, mucus, normal colon mucosa, cancer-associated stroma, tumoral epithelium, and lymphocytes[35,36].

As a majority of tumor-infiltrating lymphocytes are known to be T and B lymphocytes[37], we considered a subset of genes specifically expressed by each cell type. To define the T-cell population, we considered CD3, a transmembrane receptor glycoprotein specifically expressed at the surface of T lymphocytes. CD3 receptor is encoded by four genes: *CD3D*, *CD3E*, *CD3G*, and *CD247*[38]. We used their prediction (correlations and $p$-values in Table 1) to define the spatial localization of the T cells. Similarly, to define the B-cell population, we considered CD19 and CD20 proteins expressed exclusively by B lymphocytes[39], and used their prediction (Table 1) to define the spatial localization of the B cells (later named B-cell model).

We compared the average predicted expression of the considered genes on tiles labeled with lymphocytes and on tiles from other categories. This prediction allows to distinguish very well tiles containing lymphocytes, as measured by the area under the ROC curve (AUC): the overall AUC for distinguishing tiles with lymphocytes from all other classes is 0.94, and when considering every class separately, values range from 0.87 (lymphocytes vs adipose) to 0.98 (lymphocytes vs mucus).

We further validated the spatialization of the T-cell model on a single external H&E-CD3 double-stained slide from a LIHC sample (Fig. 4a). We calculated the correlation between the expression per tile, predicted from the H&E staining, and the actual number of T cells obtained by using QuPath software[40] on the CD3-stained slide. We obtained a correlation coefficient of $R_{\text{tile}} = 0.51$ ($p$-value $< 10^{-4}$, two-tailed Student's $t$ test). Moreover, as HE2RNA focuses particularly on histological regions associated with higher levels of gene expression, we analyzed the 100 tiles

with the highest predicted value for *CD3*-genes expression. The median number of T cells in those tiles was 36 cells, whereas the median number of T cells on all 28,123 tiles of the slide was 4, confirming the accurate spatial interpretability of the predictive model (Fig. 4c; Supplementary Fig. 6a, b).

These results show that, while trained in a weakly supervised manner based only on the overall expression of genes of TCGA samples, HE2RNA was able to distinguish lymphocytes at the level of individual tiles on an external dataset. We further assessed whether HE2RNA could distinguish different types of lymphocytes, by analyzing another slide from a different sample of LIHC and performed a H&E-CD20 double stain.

We applied both the T-cell and the B-cell models to this new slide and determined the agreement between the predicted gene expression and the number of B cells at the tile level. The T-cell model achieved a correlation coefficient $R_{\text{tile}} = 0.19$ ($p$-values $< 10^{-4}$, two-tailed Student's $t$ test), while the B-cell model achieved a significantly higher correlation $R_{\text{tile}} = 0.23$ ($p$-value $< 10^{-4}$, two-tailed Student's $t$ test). Since B cells are more sparsely distributed than T lymphocytes (with an average of 0.6 B cells per tile), an alternative metric is the ROC-AUC obtained by the model for distinguishing tiles containing more than a given number $n$ of B cells. We considered $n = 0, 1, 3$, and 11 corresponding, respectively, to the 75th, 90th, 95th, and 99th percentile of the number of B cells per tile. For every threshold except the lowest one, the B-cell model outperformed the T-cell model, with AUCs ranging from 0.66 (respectively 0.68) at $n = 0$ to 0.89 (respectively 0.81) at $n = 11$ (Fig. 4c, d and Table 2).

Conversely, we applied the B-cell model to the H&E-CD3 slide. As expected, the correlation between the prediction of the model and the number of T cells was positive ($R_{\text{tile}} = 0.39$, $p$-value $< 10^{-4}$, two-tailed Student's $t$ test), but significantly lower than the $R_{\text{tile}} = 0.51$ previously reported. Moreover, for every considered threshold, the T-cell model outperformed the B-cell model (Fig. 4e, f and Table 2). These results indicate that HE2RNA could potentially distinguish different types of lymphocytes.

In addition to genes related to immunity, we considered genes characteristic of the epithelium in prostate adenocarcinoma. Bulten et al.[41] used a similar multiple-staining procedure to generate segmentation masks of prostate epithelium of WSIs using anti-P63, CK8, and CK18 IHC stainings. The resulting dataset[42] (PESO) contains 62 H&E slides with the corresponding segmentation masks. We trained a model predicting the three corresponding genes (*TP63*, *KRT8*, and *KRT18*) on TCGA-PRAD, and applied it on this dataset. We compared the average predicted expression of those three genes at the level of individual tiles to the fraction of epithelium on those tiles, simply defined as the percentage of positive pixels of the mask. As with lymphocytes, we found a significant correlation between the prediction of HE2RNA and the fraction of epithelium ($R_{\text{tile}} = 0.41$, $p$-value $< 10^{-4}$, two-tailed Student's $t$ test) (Fig. 5a–c).

Finally, we applied a model trained to predict the expression level of *MKI67* (Table 2) to an independent dataset of 369 slides from 194 patients with LIHC[43] from hospital Henri Mondor (Mondor dataset). *MKI67* is a well-known marker of cell proliferation, expressed both by tumor and nontumor cells, clinically determined by MIB1 or Ki67 IHC staining of WSIs. Its overexpression has been found to correlate with tumor growth rate[44], histological stage[45], and tumor recurrence[46]. A higher MKI67 labeling index confers a fast progression and poor prognosis for LIHC patients[47].

Per-tile annotation of tumor vs healthy tissue was performed by a pathologist[43]. We spatialized the expression of *MKI67*, and compared it with the pathologist's annotation. First, we observed that tiles predicted to have high expression of *MKI67* were almost always located in tumoral regions: among the 10,000 (respectively 100,000) tiles with the highest predictions, 94%

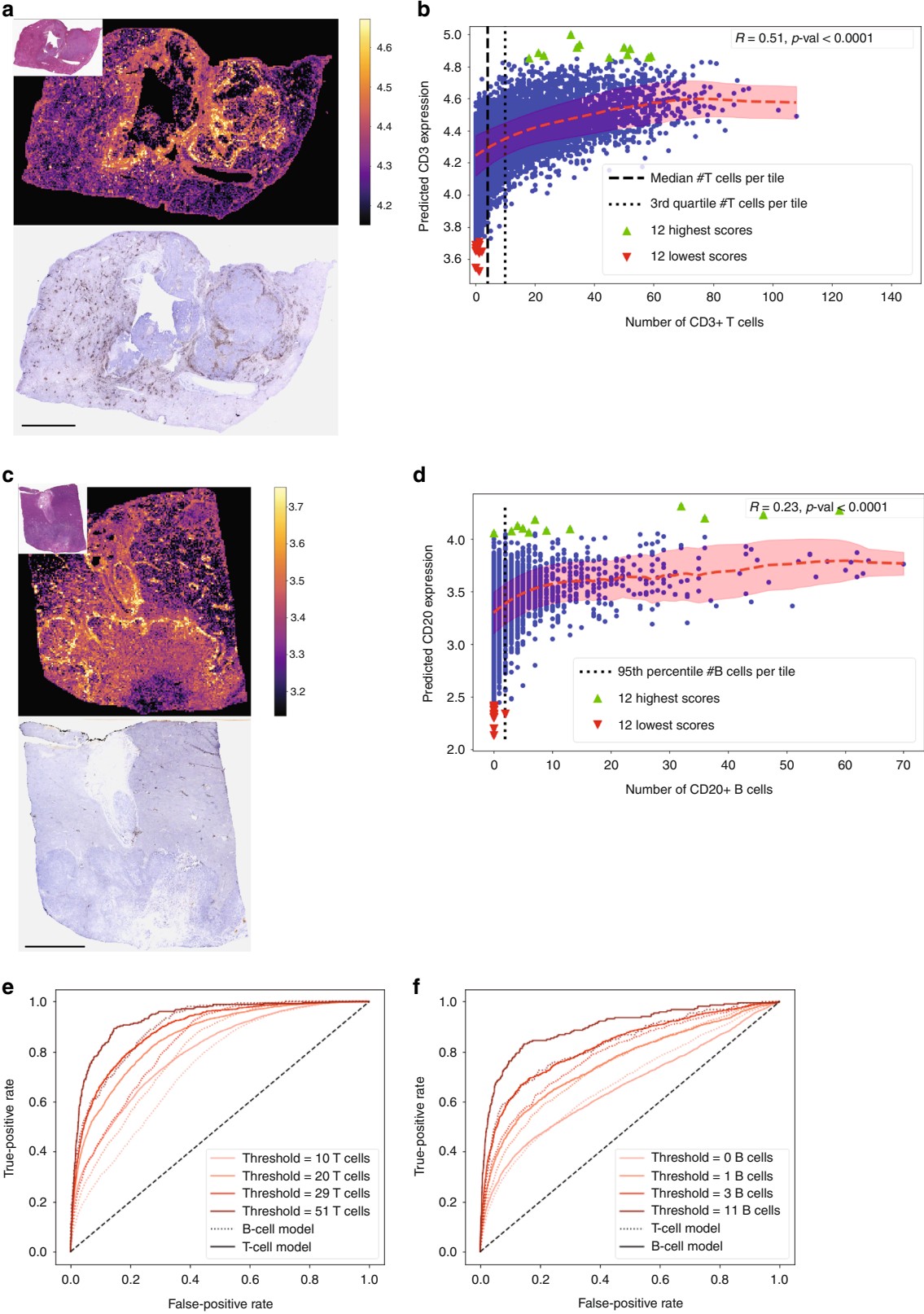

(respectively 90%) were found in the tumoral areas, while the tumor tiles only represent 57% of the whole dataset. Then, to assess how the predicted expression of *MKI67* varies within each sample, we calculated the AUC for distinguishing tiles located in the tumor from those located in the healthy tissue for every given slide. The results varied over a wide range, with an average value $AUC_{average} = 0.65$ and a median value $AUC_{median} = 0.67$, with a significant dependence on the tumor stage, measured by the Barcelona Clinic Liver Cancer (BCLC) stage[48]. For patients with an advanced-stage tumor, the predicted expression of

**Fig. 4 Virtual spatialization of *CD3* and *CD20* expression, confirmed by immunohistochemistry. a** Top left inset: H&E-stained slides were obtained from a LIHC patient. Main top image: The corresponding heatmap of the CD3-encoding genes expression predicted by our model. Main bottom image: CD3 immunohistochemistry (IHC) results obtained by washing out H&E stain and staining the same slide for IHC. **b** Pearson's coefficient ($R = 0.51$, $p$-value < $10^{-4}$, two-tailed Student's $t$ test) for the correlation between the CD3 expression predicted by our model and the percentage of CD3$^+$ cells actually detected on the IHC slide. The red dashed line indicates the average predicted expression per tile as a function of the number of CD3$^+$ cells; shaded area: s.d. The vertical dashed line indicates the median number of CD3$^+$ cells per tile, and the dotted line the 3rd quartile. **c, d** Same as in **a** and **b** using *CD19* and *CD20* coding genes and one CD20 IHC ($R = 0.23$, $p$-value < $10^{-4}$, two-tailed Student's $t$ test). Red dashed line: average prediction per tile as a function of the number of CD20$^+$ cells; shaded area: s.d. Vertical dotted line: 95th percentile of the number of CD20$^+$ cells per tile (median = 0). **e** ROC curves for distinguishing tiles from the HE/CD3 slide with a number of T cells above a given threshold (with threshold values corresponding to the 75th, 90th, 95th, and 99th percentile of the number of T cells per tile), obtained by applying both the T-cell model and the B-cell model. The dashed line is the expected ROC curve from a random classifier. **f** Same as in **e** for tiles from the HE/CD20 slide with a number of B cells above a given threshold. Scale bars: 5 mm. One slide was double-stained for each IHC marker.

**Table 2 Comparison of per-tile predictions of the T- and B-cell model to the real number of T and B cells.**

|  | T-cell model | B-cell model |
|---|---|---|
| Correlation coefficient with # T cells on the H&E/CD3-stained slide | 0.51 | 0.39 |
| AUC threshold 10 T cells | 0.79 | 0.73 |
| AUC threshold 20 T cells | 0.86 | 0.79 |
| AUC threshold 25 T cells | 0.89 | 0.83 |
| AUC threshold 51 T cells | 0.93 | 0.90 |
| Correlation coefficient with # B cells on the H&E/CD20-stained slide | 0.19 | 0.23 |
| AUC threshold 0 B cells | 0.68 | 0.66 |
| AUC threshold 3 B cells | 0.74 | 0.76 |
| AUC threshold 5 B cells | 0.78 | 0.82 |
| AUC threshold 11 B cells | 0.81 | 0.89 |

Comparaison was calculated respectively from the HE/CD3 and HE/CD20 double-stained slides. For the HE/CD3 slide (respectively HE/CD20), we display the correlation between the tile-level predictions of each model and the number of T cells (respectively B cells) per tile, as well as the AUC for distinguishing tiles that contain more T cells (respectively B cells) than a given threshold (respectively 75th, 90th, 95th, and 99th percentiles).

*MKI67* overlapped neatly with the tumor area, while for more early stages we observed less discriminate patterns: AUC$_{average}$ = 0.63 (AUC$_{median}$ = 0.65) for patients with BCLC stage A, AUC$_{average}$ = 0.63 (AUC$_{median}$ = 0.62) for patients with BCLC stage B, and AUC$_{average}$ = 0.74 (AUC$_{median}$ = 0.76) for patients with BCLC stage C (Table 3).

Finally, we also observed that the sample-wise prediction for the expression of *MKI67* is predictive of a high BCLC stage, with an AUC of 0.80 for distinguishing stage C from stages A and B. Altogether, these results are consistent with the fact that a high expression level of *MKI67* is correlated with the most advanced stages of liver cancer (Fig. 5d, e).

**HE2RNA for microsatellite instability status prediction**. HE2RNA provides a transcriptomic representation of WSIs, of potential utility for different clinical situations. This representation is obtained while learning the transcriptome, when HE2RNA transforms each WSI into a vector of P features corresponding to the dimensionality of the last hidden representation of the neural network. To illustrate the clinical applications of this representation, we studied microsatellite instability (MSI) status prediction as a diagnosis use case directly using WSIs. MSI phenotype is frequently observed in adrenocortical, rectal, colon, stomach, and endometrial tumors[49], and also occur in other cancers (e.g., breast, prostate)[50]. The failure to correct replication errors at tandem repeats of short DNA sequences known as microsatellites can lead to the phenomenon of high-level MSI

(MSI-H)[51,52], recently identified as a predictor of the efficacy of immunotherapy[53,54]. The analysis of gene expression prediction, restricted to MSI-H patients from TCGA-COAD (81 samples), revealed that a surprisingly high number of genes were significantly well-predicted by HE2RNA on this subset (1027 genes well-predicted under HS correction), more than on the whole dataset (324 well-predicted genes for 463 samples) or on the subset of MSS patients (592 well-predicted genes for 277 samples) (Fig. 6a). A gene set enrichment analysis of the genes well-predicted in MSI-H patients revealed an enrichment in T-cell activation and immune activation (PD-1 signaling, interferon gamma signaling…). These results confirmed the high performance of HE2RNA to predict immune infiltrate and are aligned with the known higher immune infiltration observed in these patients and linked to their positive response to immunotherapies (Fig. 6b). Performing a similar analysis in MSS patients, we identified mostly pathways involved in RNA metabolism and translation regulation (formation of free 40S subunits, translation initiation…) (Supplementary Fig. 7).

MSI status can be determined by immunohistochemistry (IHC) or genetic analyses[55]. Although broadly recommended by medical guidelines, systematic screening for MSI condition is performed at high-volume tertiary care centers, but less frequently in low-income hospitals. There is, therefore, a great need to screen directly WSIs from cancer patients with a high probability of MSI-H to facilitate access to immunotherapy. A recent study[56] showed that CNNs can learn to predict MSI status directly from histology slides for stomach adenocarcinoma and colorectal cancer. Based on these results, we collected RNA-Seq measurements, WSIs, and MSI status of each patient from TCGA colorectal cancer dataset, later referred to as TCGA-CRC-DX (corresponding to TCGA-COAD and TCGA-READ) and TCGA-STAD dataset, and investigated the effects of integrating RNA-Seq information for the prediction of MSI status from pathology images. To enable the comparison with previously published results[56], in addition to the formalin-fixed, paraffin-embedded (FFPE) slides used above, we also collected frozen slides from colorectal cancer cases (TCGA-CRC-KR). Patients with MSI-High (MSI-H) and microsatellite stable (MSS) were both randomly split between two hospitals named A and B. In hospital A, the first subset of patients was used to train a simplified version of HE2RNA (see "Methods") to predict RNA-Seq data, but not MSI status (Fig. 6c). In hospital B, the second subset of patients was used to directly train a binary classifier MSI-H vs. MSS. To test HE2RNA's robustness, we considered several splitting proportions for the data subsets used in the two hospitals, ranging from 5/6–1/6 (corresponding to 388–78 patients for the TCGA-CRC-DX dataset), to 100% of the patients in hospital B. Two different performance patterns emerged as shown on Fig. 6d (TCGA-CRC-DX) and Supplementary Fig. 8 (TCGA-STAD and TCGA-CRC-KR). When a few samples were used to learn the

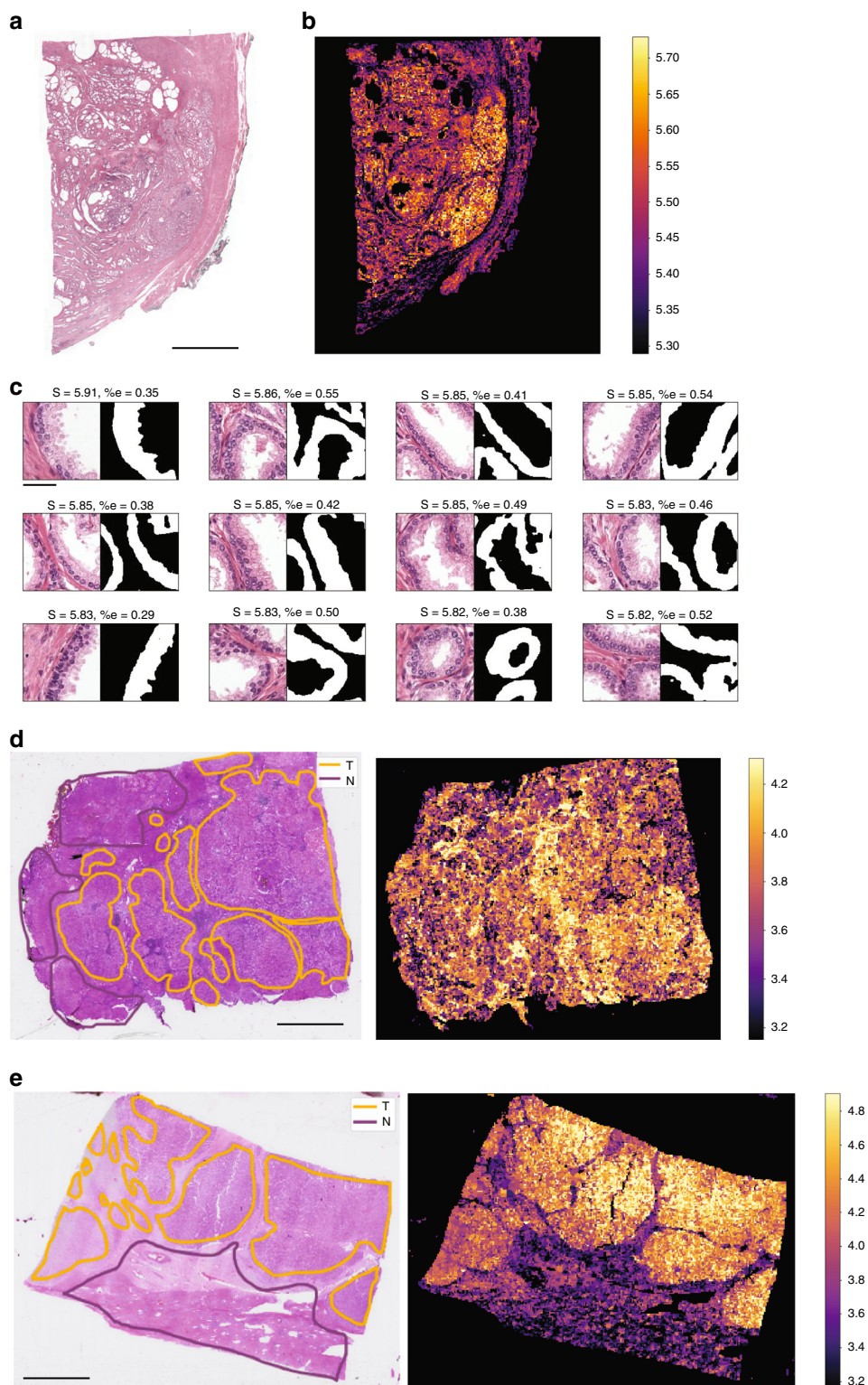

**Fig. 5 Virtual spatialization of epithelium-associated genes (*TP63, KRT8*, and *KRT18*) and MKI67 expression. a** Representative H&E slide from the PESO dataset[59] (*n* = 62 slides with segmentation mask). **b**. Heatmap for the expression of *TP63, KRT8*, and *KRT18* predicted by HE2RNA for the slide in **a**. **c** Tiles with highest predicted expression for those genes on this slide, with the segmentation mask of epithelium obtained from an IHC staining of the same slide[42]. S = Score corresponding to the log expression score of each tile; %e = fraction of pixels marked as belonging to the epithelium (*n* = 21,714 tiles in total). **d** Spatialization of *MKI67* predicted expression on a liver hepatocarcinoma sample from an early stage tumor (BCLC stage A) (*n* = 284 samples)[43]. Left panel: Representative H&E staining, with annotation of tumor (T) and nontumor (N) areas performed by a pathologist. Right panel: Heatmap for the expression of *MKI67* predicted by the model. **e** Same as **d**, for a sample from an advanced tumor (BCLC stage C) (*n* = 65 samples). In **a**, **d**, **e**, scale bar: 5 mm. In **c**, scale bar: 100 μm. **d**, **e** are representative slides (*n* = 369 annotated slides). BCLC Barcelona Clinic Liver Cancer.

**Table 3 Agreement between the predicted expression of *MKI67* and the tumor annotation.**

| BCLC stage | # Patients | # Slides | # Tiles healthy tissue | # Tiles tumor | AUC mean | AUC median |
|---|---|---|---|---|---|---|
| A | 147 | 284 | 1.85 M | 2.46 M | 0.63 | 0.65 |
| B | 4 | 20 | 175 k | 122 k | 0.63 | 0.62 |
| C | 35 | 65 | 325 k | 570 k | 0.74 | 0.76 |

*BCLC* Barcelona Clinic Liver Cancer, # number.
The metric is the ROC-AUC per slide for determining whether a tile is located in the tumor or in healthy tissue.

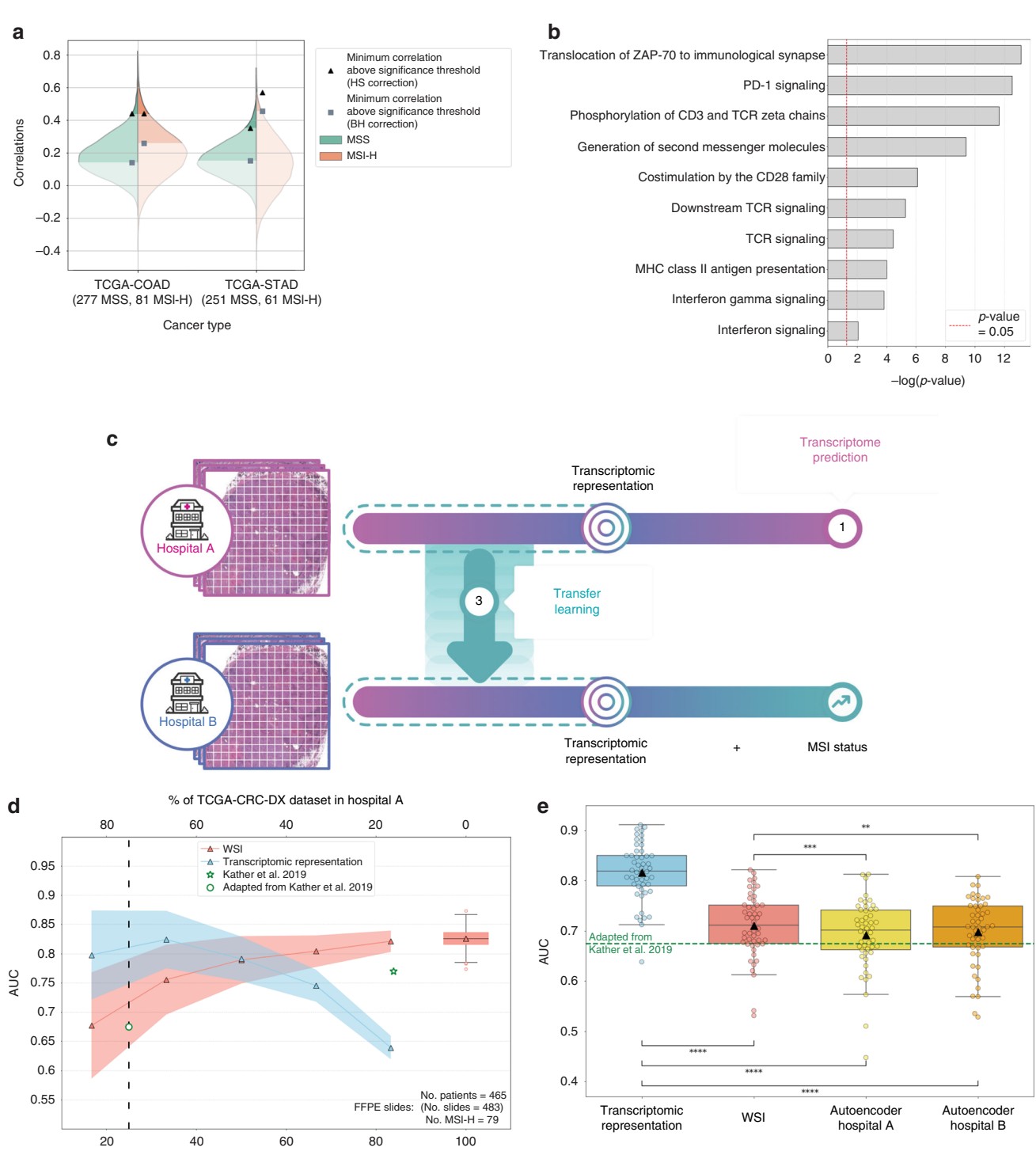

**Fig. 6 Prediction of microsatellite instability status using transfer learning from transcriptomic representation. a** Distribution of Pearson correlation coefficients on TCGA-COAD and TCGA-STAD, for microsatellite-stable (MSS) patients (green) or patients with high-level microsatellite instability (MSI-H) (orange). Black triangles (respectively grey squares): minimum correlation required for significance under Holm–Šidák (respectively Benjamini–Hochberg) correction. **b** Computational analysis of the most accurately predicted genes in MSI-H patients from TCGA-COAD. p-values were calculated using right-tailed Fisher's exact test. **c** Setup: in hospital A, a neural network is trained to predict gene expression from WSIs. The internal transcriptomic representation is then used in hospital B to improve MSI status prediction. **d** Area under the ROC curve (AUC) for the model based on the transcriptomic representation (blue) or directly based on WSIs (red), as a function of the fraction of the TGCA-CRC-DX dataset used in the two hospitals ($n = 50$ data splits per fraction, averaged over ten different three-fold cross-validations (CVs); solid line and triangles: mean over splits; shaded area: s.d.). Boxplot: distribution of AUCs (500 three-fold CVs) over the whole dataset, for the model based on WSIs (box: interquartile range (IQR); horizontal line: median; whiskers: 1.5 times IQR, triangle: mean; circles: outliers). Star: result from Kather et al.[56]; its location accounts for the different number of patients in the training set with respect to this manuscript; circle: result from the same method[56] with 25% of the data in hospital B (see "Methods"). Dashed line: fractions of the dataset used in panel c. **e** Boxplots (defined as in panel **d**) of the distribution of AUCs for MSI status classifiers at hospital B, trained, respectively, on the 256-dimensional transcriptomic representation, WSIs, and 256-dimensional representations given by two autoencoders trained on hospital A and B subsets. Dashed line: result obtained by adapting Kather et al.[56] method. Circles: average over ten three-fold CVs for each split between the hospitals ($n = 50$). **p < 0.01, ***p < 0.001, and ****p < 0.0001, two-tailed Wilcoxon test. WSI whole-slide image. Hospital illustration based on "hospital" by H Alberto Gongora, from thenounproject.com, used under CC-BY 3.0/colored.

**Table 4 ROC-AUCs for the prediction of MSI status in two extreme regimes.**

| Regime | Dataset | Transcriptomic (this paper) | WSIs (this paper) | WSIs. (ref. [56]) |
|---|---|---|---|---|
| 1 (100% of data in hospital B) | TCGA-CRC-DX | — | 0.82 | 0.77 |
| 1 | TCGA-CRC-KR | — | 0.83 | 0.84 |
| 1 | TCGA-STAD | — | 0.76 | 0.81 |
| 2 (25% of data in hospital B) | TCGA-CRC-DX | 0.81 | 0.71 | 0.68 |
| 2 | TCGA-CRC-KR | 0.79 | 0.72 | 0.63 |
| 2 | TCGA-STAD | 0.66 | 0.63 | 0.65 |

In regime 1, 100% of the data in hospital B. In regime 2, 75% of the data in hospital A and 25% in hospital B. TCGA-CRC-DX: TCGA colorectal cancer dataset (FFPE slides), TCGA-CRC-KR: TCGA colorectal cancer dataset (frozen slides) and TCGA-STAD: Stomach adenocarcinoma.

transcriptomic representation (hospital A) and most patients were used to train the model for MSI prediction, the classifier trained directly on WSI outperformed the classifier using the WSI transcriptomic representation learned at hospital A. With 100% of the data in hospital B, the classifier trained directly on WSIs demonstrated similar results than previous studies[56] (Table 4). In contrast, when a few examples were available to train the classifiers at hospital B, a reverse pattern was observed. With a 3/4–1/4 data split between the two hospitals, the transcriptomic-based model gave more accurate predictions than the WSI-based model (two-tailed Wilcoxon test: $p < 0.0001$; Fig. 6e). To confirm that the performance improvement was not only due to the dimensionality of the representation, we also considered the same MSI classifier trained with a 256-dimensional representation (same size as our transcriptomic representation) given by two different autoencoders, respectively trained on hospital A and hospital B subsets. A two-tailed Wilcoxon test confirmed ($p$-value < 0.0001) that the classifier based on the transcriptomic representation of WSI slides still outperformed the direct classifiers with reduced dimension representations. Finally, to compare HE2RNA with the work of Kather et al.[56], we re-implemented the published method in the 3/4–1/4 setting (see "Methods"). The use of the transcriptomic representation resulted in a significant improvement on TCGA-CRC-DX ($AUC_{HE2RNA} = 0.81$ vs $AUC_{Kather} = 0.68$) and TCGA-CRC-KR (respectively, 0.79 and 0.63) datasets. On TCGA-STAD, the methods showed close performances (respectively, 0.66 and 0.65) (Table 4). Similar results were obtained when MSI-low patients were included together with the MSS patients in a MSI-non high (MSI-NH) class (Supplementary Fig. 8).

Altogether, our results demonstrated that learning and transferring a transcriptomic representation learned on a dataset with such available data could help increase the prediction performances of WSI-based models for a clinical diagnosis purpose such as MSI detection in hospitals where transcriptomic and molecular profiling are not done routinely.

## Discussion

We presented HE2RNA, a deep-learning model to infer transcriptomic profiles from histological images. This paper covers a broad spectrum of applications using the transcriptomic information encoded within cancer tissue images. HE2RNA robustly and consistently predicted subsets of genes expressed in different cancer types, including genes involved in immune cell activation status and immune cell signaling. We hypothesize that the algorithm can recognize immune cells and correlate their presence with the expression of a subset of protein-coding genes (such as CQ1B). Major breakthroughs in cancer therapy are driven by discoveries of treatment targets for immunotherapy in many types of cancer. HE2RNA could be a useful tool for pathologists and oncologists to consistently quantify immune infiltration or guide treatment decision in the context of immunotherapy. In the future, it would be interesting to determine whether similar models could be trained to predict patient response to immunotherapy, making it possible to identify histology-based biomarkers of treatment response.

HE2RNA also correctly predicts the expression of genes involved in cancer type-specific pathways, such as fibrosis in hepatocellular carcinoma, or CHK gene expression in breast cancer. The ability of our model to detect molecular and cellular modifications within cancer cells was also confirmed by the greater prediction accuracy for defined gene signatures than for lists of random genes. Many previous studies have shown that tissue sections contain tremendous amounts of information[57–59]. Our model might capture more subtle structures in tissue images, unraveling interesting histological patterns, an enlightening

important tumor region for the development of specific cancer types. Strikingly, even with RNA-Seq profiles obtained from bulk cells isolated from WSIs, our model was able to predict the spatial expression of various genes from the H&E slide alone. In particular, HE2RNA was able to spatialize differentially genes specifically expressed by T cells or B cells, even though discriminating between those two types from their morphology alone is notoriously difficult, as indicated by scanning-flow cytometry experiments[50]. Such methods could be extended to other genes, including genes related to immunity, the expression levels of which were well-predicted by our model, and which could represent a major tool for medical diagnosis and prediction, by providing virtual multiplexed staining for all WSIs alone. These innovative approaches may help to overcome technical issues in IHC, such as fixation or antigen retrieval, together with the high level of variability between observers. However, recent studies[60] have shown that mRNA and protein expressions may be poorly correlated due to various factors, such as different half-lives and post-transcription machinery. Thus, a joint analysis of the transcriptomic and proteomic data could provide useful insights and increase the overall performance of our model.

CNNs for image recognition make use of an internal representation of the original data that they infer. The features of this latent space encode the statistics of natural images and the information of importance for image recognition. Similarly, the internal transcriptomic representation, learned by HE2RNA during the prediction of RNA-Seq data, may constitute an important step toward understanding the biological descriptors required for clinical classification problems and the link between the information contained at the tissue and molecular levels. We have shown that the lower-dimensional transcriptomic representation learned during the RNA-Seq prediction task can be very powerful when transferred to other datasets used for a different task. This seems to be particularly true for small WSI datasets, for which even partial information about the connection between histological and molecular information can significantly improve the performance of deep-learning models. Such situations are frequently encountered when a research center develops models for predicting treatment outcomes from a small dataset, with access to additional external data but not the corresponding biological status of interest. The approach proposed here could constitute a paradigm in transfer learning in medicine. We used MSI status prediction from a small dataset of H&E-stained images as a representative case study, and we showed that the use of a transcriptomic representation in a transfer learning framework outperformed similar models based on less informative representations, such as WSI only. This aspect is particularly important, as MSI-H status is a predictive biomarker of response to immunotherapy[53,54], and of better overall prognosis relative to patients with microsatellite-stable disease (MSS), in gastric adenocarcinoma and CRC[55]. However, not all patients are screened for MSI status outside of high-volume tertiary care centers. Our model could therefore be used in the future to facilitate the definition of patient MSI status and to facilitate access to immunotherapies for a larger number of eligible patients. The practical implications of predicting gene expression level from H&E slides should not be underestimated. In the future, the performance of this model may improve considerably, through the use of larger, richer datasets for training. Such approaches could also be used to detect histological subtypes, genetic mutations, to map the infiltration of tumors by tumor-infiltrating lymphocytes (TILs), or to predict molecular profiling from pathology slides, such as the hormonal status of breast cancer cases. HE2RNA offers useful applications to enable more patients to gain access to precision medicine, without the need for hospitals to resort to NGS techniques.

## Methods

**TCGA Pan-cancer dataset**. This study was based on publicly available data from TCGA (https://portal.gdc.cancer.gov/). We selected samples from primary tumors only, for which both RNA-Seq and WSI data were available. Transcriptomic data (FPKM-UQ) were extracted from frozen tissues, and the slides analyzed were digitized H&E-stained formalin-fixed, paraffin-embedded (FFPE) histology slides, referred to here as whole-slide images (WSIs). WSIs were available for the cancer types (and corresponding abbreviations) listed in Supplementary Table 1.

**Gene expression pre-filtering and normalization**. Gene expression data in fragment per kilobase million with upper-quartile normalization (FPKM-UQ) were available for 60,483 Ensembl gene identifiers, many corresponding to noncoding genomic regions. We chose to exclude genes with a median expression of zero (i.e., not expressed in more than half the samples considered), to improve the interpretability of the results, using an approach similar to that used in existing tools[61] for differential gene expression analysis. After the application of this filter, 30,839 genes remained, 17,759 of which encoded proteins (all Ensembl genes associated with a corresponding Ensembl protein ID, for the Hg19 human genome sequence). Our selection included almost 90% of known protein-coding genes. Gene expressions values covered several orders of magnitude. Thus, regression analysis directly on raw RNA-Seq data would lead the model to focus only on the most strongly expressed genes, which would dominate the mean squared error. We overcame this problem by an $a \rightarrow \log_{10}(1 + a)$ transformation on gene FPKM-UQ expression values.

**Preprocessing of whole-slide images**. The application of deep-learning algorithms to histological data is a challenging problem, particularly due to the high dimensionality of the data (up to $100,000 \times 100,000$ pixels for a single whole-slide image) and the small size of available datasets. We divided the whole-slide images into squares of $112 \times 112$ μm ($224 \times 224$ pixels) called "tiles", and used the Otsu algorithm[62] (as implemented in python package skimage) to select only those containing tissue, excluding the white background. We sampled a maximum of 8000 such tiles from each slide. We then extracted 2048 features from those tiles with a 50-layer ResNet[63] pretrained on the ImageNet dataset[64] (using the Keras implementation), such that a slide could be represented as a $8000 \times 2048$ matrix.

For the first phase of this work (transcriptome prediction), we accelerated the training of our models through a simple preprocessing step inspired by simple linear iterative clustering (SLIC)[65]: we used the k-means algorithm (as implemented in python package libkmcuda) to create 100 clusters (supertiles) of tiles on the basis of tile location on the slide, and we averaged the features of the tiles within each cluster. The use of these supertiles reduces the dimensions of a slide to $100 \times 2048$. The model was first trained on this reduced dataset, with all the TCGA data. Then, for specific organs, fine-tuning was achieved with full-scale data from the organs concerned only.

**HE2RNA model architecture**. The HE2RNA model is a multilayer perceptron (MLP), applied to every tile (or super-tile) of the slide. This choice, as opposed to a simple linear regression, allows to perform multitask learning by taking into account the correlations between multiple gene expressions at the (super-)tile level. For practical purposes, this is equivalent to applying successive 1D convolutions of kernel size 1 and stride 1 to slide data. The activation function is a rectified linear unit and dropout is applied between consecutive layers. For an input matrix of size $n_{tiles} \times 2048$ (with $n_{tiles} = 100$ or $n_{tiles} = 8000$), the output of this neural network is a matrix of size $n_{tiles} \times n_{genes}$, where $n_{genes}$ is the number of genes for which expression is to be predicted. Thus, the model produces one prediction per gene and per tile, but the real value is available only at the scale of the slide. For this reason, tile predictions must be aggregated to produce one prediction per gene and per slide for model training and the calculation of metrics.

The aggregation strategy we use is the following. During the training phase, the model randomly samples a number $k$ at every iteration, and for every gene, averages only the $k$ highest tile predictions to produce the slide-level prediction. Indeed, we are predicting the logarithm of gene expression, and the (super-)tiles with the highest levels of expression should contribute the most to this value. The number $k$ is sampled from the list $L = (1, 2, 5, 10, 20, 50, 100)$ for super-tile-preprocessed data ($n_{tiles} = 100$), and from the list $L = (10, 20, 50, 100, 200, 500, 1000, 2000, 5000)$ for full-scale data ($n_{tiles} = 8000$). For instance, if $k = 1$, the slide-level prediction for a given gene is the largest (super-)tile prediction for this gene, whereas if $k = n_{tiles}$, the slide-level prediction is the average of all tile predictions. More generally, the slide-level prediction $S$ for a given gene is defined as follows,

$$S(k) = \sum_{i=1}^{k} s_i/k,$$

where the $s_i$ are the tile-wise predictions for this gene, ordered from the largest ($s_1$) to the smallest. This stochastic aggregation increases the difficulty of the task and thus reduces overfitting. As such, it acts as a regularization process. During inference, slide-level predictions obtained with every possible value of $k$ are

averaged,

$$S = \sum_{k \in L} S(k)/|L|.$$

This is equivalent to calculating a weighted mean of per-tile predictions, with a greater weight for tiles for which the model predicts high levels of expression. A summary of the weights of tile predictions in this weighted average as a function of their rank is detailed in Supplementary Table S2.

**HE2RNA training and evaluation.** HE2RNA was trained with a five-fold cross-validation designed to meet the following requirements: every sample from a patient should be in the same fold, and, when training on all TCGA data, TCGA projects should be evenly distributed between folds. When training a model on a single subset (e.g., BRCA), we ensured that the folds used for cross-validation were consistent with those used for all the TCGA data. The model simultaneously predicting all genes for all types of cancers was trained on all TCGA data (10,514 samples), with super-tile preprocessing for a maximum of 200 training epochs. Training was stopped when the average per-gene and per-cancer type correlation computed on a validation set comprising 10% of the training data did not improve for 50 consecutive epochs.

When specific sets of genes (such as CD3 receptor encoding genes) were used to produce a precise heatmap of gene expression, the model was trained at the finest scale.

To optimize the trade-off between achieving optimal performance using full-scale TCGA data (10,514 slides × 8000 tiles × 2048 features) and minimizing the machine time for training, we first trained the model for 200 epochs on all super-tile-preprocessed TGCA data (10,514 slides × 100 tiles × 2048 features), before continuing training for 100 epochs on full-scale data for a subset of five organs of interest (3464 slides × 8000 tiles × 2048 features). The organs of interest we chose for training the models of part 2 are those on which we performed spatialization (colon, liver, and prostate), augmented with the two largest TCGA datasets (lung and breast), which are also those with the largest numbers of significantly well-predicted genes. In Table 1, the performance metrics for the genes of interest for virtual spatialization are displayed.

The performance metric was always calculated separately for each organ considered, to prevent bias. Gene expression levels can differ considerably between organs and a good performance for gene expression could otherwise be achieved simply through recognition of the organ of origin.

All models were trained with the Adam optimizer and a learning rate of $3 \times 10^{-4}$. We used a minibatch size of 16 when working with super-tile preprocessing, and 4 when working on full-scale data due to GPU memory constraints.

**Per-tile inference.** As mentioned above, the model produces one score per tile and per gene. For each gene, the predictions per tile are subsequently aggregated to produce one prediction per gene at the level of the slide. To generate virtual spatialization maps, we simply omit this aggregation step and interpret the score of a tile as the predicted gene expression for this tile.

**Gene signatures for gene set enrichment analyses.** Using Gene Set Enrichment Analysis (GSEA) software and Molecular Signatures Database v6.2, we analyzed the following signatures for the following pathways (Supplementary Table 3). For each pathway, we selected the genes present in at least two of the chosen signatures (see Supplementary Table 7 for a complete list of the genes retained for each pathway).

**Virtual spatialization and double staining.** Features were extracted from every tile containing matter, with the ResNet50 algorithm. The model was then used to calculate a score for each tile. Finally, the heatmap at the scale of the whole slide was obtained by weighting each tile by its score. For the single double-stained HE-CD3 (respectively HE-CD20) slide, we extracted a total of 28,123 (respectively 25,088) tiles to generate the heatmap shown in Fig. 4.

The tissue section was first stained with hematein, eosin, and saffron, coverslipped and scanned using a Leica Aperio Scanner. The coverslip and the mounting reagents were removed using acetone, and the slide was further unstained using an alcohol/acid solution (1%). Immunostaining was performed using a Leica Bondmax Autostainer (Leica Biosystems, Wetzlar, Germany) with an anti-CD3 antibody (Dako, Santa Clara, California, Clone, F7.2.38, dilution 1/50), or an anti-CD20 antibody (Dako, Santa Clara, Clone L26, dilution 1/50) according to manufacturer's instructions. Antigen retrieval was performed with E2 reagent, and Immunodetection was performed using Polymer and 3,3' di-aminobenzidine (Bond Polymer Refine Detection Kit, Leica Biosystems). The immunostained slide was further scanned.

For comparison of the predicted heatmap and the CD3- or CD20-stained WSI, we used QuPath software on the WSIs to estimate the actual number of T cells per tile, and then calculated the correlation between this number and the score per tile.

**Data for virtual spatialization of MKI67.** The Mondor dataset consists of 390 slides (NDPI format, 40x magnification) from 206 patients, with the following inclusion criteria:

Patients treated by surgical resection without any prior anti-tumor therapy
Available follow-up
Unequivocal diagnosis of HCC
Available histological slides from formalin-fixed paraffin-embedded material
Lack of extra-hepatic metastatic disease at time of surgery.

The tumor and nontumor areas were annotated by an expert liver pathologist (JC) on 369 slides (corresponding to 190 patients). BCLC stage was retrieved from medical records. Informed consent was obtained for each patient, and the study was approved by Saint Louis Hospital Ethics Committee (Assistance Publique—Hôpitaux de Paris).

**Data for MSI prediction.** We used the histology images for $n = 465$ patients with colorectal carcinoma (TCGA-COAD and TCGA-READ) (diagnostic slides, FFPE tissue) from the TCGA dataset, together with the corresponding MSI status data obtained from TCGAbiolinks[66]. MSI status is assessed by measuring the lengths of a set of mono- and dinucleotide repeats in tumor and matched normal DNA. The MSI-Mono-Dinucleotide Assay used by the Cancer Genome Atlas (TCGA) consists of a panel of four mononucleotide-repeat loci (polyadenine tracts BAT25, BAT26, BAT40, and transforming growth factor receptor type II) and three dinucleotide repeat loci (CA repeats in D2S123, D5S346, and D17S250)[67,68]. Tumors are classified as MSI-H if more than 40% of the markers are altered, MSI-low (MSI-L) if less than 40% of the markers are altered and MSS if no marker is altered[69].

**Transcriptomic learning for MSI prediction.** For transcriptomic learning at hospital A, we used the simplest version of HE2RNA, in which the input for each slide was the mean value over the ResNet50 representation of every tile, which is equivalent to a preprocessing with one super-tile (see Supplementary Fig. 1).

We also set the MLP architecture as follows: two hidden dense layers of 1024 and 256 neurons with sigmoid activation followed by the last prediction layer with 28,334 output neurons and linear activation (28,334 being the number of coding or noncoding genes with nonzero median expression levels over the samples of CRC dataset). The model was trained for 50 epochs without hyperparameter tuning. For the MSI classifier at hospital B, we compared two different models. The first one was an MLP with two hidden dense layers of 256 and 128 neurons and a one-neuron output layer, all with sigmoid activation, also fed with the all-image average of the ResNet50 representation of the tiles and trained for 50 epochs. The second consisted of a neural network with the same architecture, without the first hidden layer of 256 neurons; this second classifier was trained on the 256-dimensional representations of hospital B WSIs, given by the transcriptomic representation learned at hospital A, or by an autoencoder (with a mirror architecture and three hidden dense layers of 1024, 256, and 1024 neurons, with rectified linear activation unit (ReLU)[70] and linear activations), trained on either the hospital A or hospital B subset. We performed different experiments, for different ratios of sample size between the hospital A and B subsets. We performed a three-fold cross-validation on the hospital A task, and the transcriptomic representation for the hospital B dataset was obtained by averaging the three corresponding inferences for the hospital B subset. MSI status was also predicted in a three-fold cross-validation setup. Moreover, for each A-/B-subset size ratio, the random split between the two hospitals was bootstrapped 50 times to generate robust performance estimates. Whenever several samples were available for the same patient, we simply averaged the predictions of the model over every sample, to obtain one single prediction for the patient.

For comparison with the work of Kather et al.[56], we re-implemented the method described in this study, in the regime where the hospital B dataset represents 1/4 of the total amount of data (dashed line on Fig. 6). For instance, for the TCGA-CRC FFPE dataset, we considered the configuration where there are 116 patients with MSI status in hospital B. We fine tuned a ResNet18, pretrained on the ImageNet database, on tile images with all layers but the ten last ones frozen, with the ADAM optimizer, and a learning rate of $10^{-6}$, with a L2-penalty of $10^{-4}$ on weights. The AUC on a hold-out validation set containing 12.5% of the training data was computed every 256 iterations, and training was interrupted when this score did not improve for a given number of consecutive epochs (patience). We performed experiments with a patience of three epochs, as described in ref. [56], and with a patience of ten epochs, which slightly improves the results in this particular setting with few training samples.

**Statistics and reproducibility.** We determined whether the correlation between the prediction of the RNA-Seq expression levels for a given gene and the real values was statistically significant, by comparison with the distribution of correlations predicted by a model with the same architecture as HE2RNA but untrained. The estimated $p$-values for tests against the null hypothesis of random correlations were then corrected by the Holm–Šidák method, to account for multiple comparisons. This correction was performed separately for each cancer type. A gene was considered significantly well-predicted, for a given cancer type, if its corrected $p$-value was below 0.05. The IPA analysis in Figs. 1c–e is based on Fisher's exact tests. The direction of the change in gene expression is not taken into account in this calculation. For the analysis on LIHC and BRCA, given the large number of genes with $p$-value below 0.05, we focused on genes for which the coefficient of the

correlation between the predicted and true value was greater than 0.4: 786 genes for BRCA and 765 for LIHC.

For the analysis of the hallmark of cancer pathways, we selected, for each pathway and cancer type, 10,000 different random lists of genes of the same length as the pathway list. We then determined the correlation for all genes and the number of genes for which expression was well-predicted (in terms of the Holm–Šidák corrected $p$-value), for all these lists.

We compared the mean correlation $R_p$ over the pathway gene list, with the distribution of the mean correlation over the random gene lists and calculated the associated $p$-value. We plotted $R_p$ against the mean $R_0$ over all the 10,000 averages of $R_r$ for the random list (Fig. 3). We considered a given pathway for a given cancer type to be better predicted than a corresponding random set, when the probability $p$ of $R_r > R_p$ was <0.05.

We adopted the same approach for the percentage of well-predicted genes as a measure, but compared the percentages $f_p$ of well-predicted genes in the pathway, with the distribution $f_0$ over the 10,000 random lists.

The MSI prediction scores are expressed as area under the ROC curve (AUC) and a two-tailed Wilcoxon test was used to compare the different distributions of scores.

The analyses in this work were carried with python (version 3.7.4) and made use of the following packages: cffi (version 1.14.0), colorcorrect (version 0.9), cryptography (version 2.8), Cython (version 0.29.14), decorator (version 4.3.0), gdc-client (from https://github.com/NCI-GDC/gdc-client/tree/1c69ed0c2bfa3c6b3784bca1ab6feaed7a81f6cb), h5py (version 2.9.0), intervaltree (version 3.0.2), ipykernel (version 5.0.0), ipython (version 7.0.1), ipython-genutils (version 0.2.0), ipywidgets (version 7.4.2), jsonschema (version 2.6.0), jupyter (version 1.0.0), jupyter-client (version 5.2.3), jupyter-console (version 5.2.0), jupyter-contrib-core (version 0.3.3), jupyter-contrib-nbextensions (version 0.5.0), jupyter-core (version 4.4.0), jupyter-highlight-selected-word (version 0.2.0), jupyter-nbextensions-configurator (version 0.4.0), Keras (version 2.2.4), Keras-Applications (version 1.0.6), Keras-Preprocessing (version 1.0.5), libkmcuda (from https://github.com/src-d/kmcuda.git#subdirectory=src), lxml (version 4.4.2), matplotlib (version 3.1.1), mygene (version 3.0.0), ndg-httpsclient (version 0.5.0), numba (version 0.45.1), numpy (version 1.17.0), openslide-python (version 1.1.1), pandas (version 0.23.4), pathlib (version 1.0.1), Pillow (version 6.1.0), progressbar2 (version 3.43.1), pyasn1 (version 0.4.3), pyOpenSSL (version 18.0.0), PyYAML (version 3.13), requests (version 2.22.0), scikit-learn (version 0.21.2), scipy (version 1.2.1), seaborn (version 0.9.0), setuptools (version 45.3.0), skimage (version 0.16.2), statsmodels (version 0.9.0), tables (version 3.5.2), tensorboard (version 1.14.0), tensorboardX (version 1.4), tensorflow-estimator (version 1.14.0), tensorflow-gpu (version 1.14.0), termcolor (version 1.1.0), torch (version 1.4.0), torchvision (version 0.5.0), tqdm (version 4.32.2).

**Reporting summary**. Further information on research design is available in the Nature Research Reporting Summary linked to this article.

## Data availability
The TCGA dataset is publicly available at the TCGA portal (https://portal.gdc.cancer.gov). Labeled tiles from colorectal cancer samples "100,000 histological images of human colorectal cancer and healthy tissue" and the PESO dataset are publicly available at https://zenodo.org/record/1214456#.XpWJbm46—w and https://doi.org/10.5281/zenodo.1485967, respectively. The Mondor dataset (whole-slide images) is available from hospital Henri Mondor but restrictions apply to the availability of data, which were used with permission for the current study, and so are not publicly available. The data, or a test subset, may be available from hospital Henri Mondor subject to ethical approvals (details of the data and how to request access are available from Dr Julien Calderado at Hospital Henri Mondor). Model interpretability can be explored at: https://owkin.com/he2rna-result-visualization/.

## Code availability
Source code is available at https://github.com/owkin/HE2RNA_code.

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

## Acknowledgements

We thank The Cancer Genome Atlas (TCGA) program for providing free access to histology slide images and RNA-Seq data, and Qiagen Bioinformatics for allowing us to use their Ingenuity Pathway Analysis (IPA) software on free trial. We thank Simon Jégou, Charles Maussion, and Eric Tramel for insightful discussions.

## Author contributions

B.S., J.C., M.S., M.Z., T.C., P.C., and G.W. designed the experiments; P.M. performed the double-staining experiments; B.S., A.R., C.S., and P.C. performed the numerical experiments and wrote the code to achieve the different tasks; B.S., A.R., E.P., A.K., M.M., and P.C., analyzed the results. B.S., A.R., E.P., S.T., M.M., P.C., and G.W. wrote the paper with the assistance and feedback of all the other co-authors.

## Competing interests

The authors declare the following competing interests: Employment: B.S., A.R., E.P., C.S., A.K., M.S., S.T., M.Z., T.C, M.M., P.C., G.W. are employed by Owkin, Inc. Advisory: J.C. reports consulting fees at Owkin, Inc. The remaining authors declare no competing interests.
