## [Peer Review File · Nature Communications]

Reviewers' comments:

Reviewer #1 (Remarks to the Author):

In this manuscript the authors provide an approach to use of machine learning for predicting RNA-seq profiles from histopathology images. There is good clinical rationale for such research to identify predictive/prognostic molecular subtypes from standard H&E stained slides and potentially avoid more expensive molecular tests in certain situations. The strengths of the study are that a large TCGA dataset is used including both training and validation cohorts. A potential limitation is that since majority of the data has been developed from TCGA data there may be inherent artefacts that limit the applicability to other datasets. Ideally validation using a non-TCGA dataset should also be performed.

It is noted however that validation using CD3 double-stain on a liver hepatocellular carcinoma series is provided, however details on this independent dataset should be more clearly provided i.e. number of tumours, number of slides and how the processing was performed. This validation approach using staining for the T cell marker CD3 and validating that number of T-cells present in a tile correlates with expression of CD3 RNA seems a good validation step. However, whether the algorithm is equally successful at identifying tumour specific molecular alterations in individual image tiles is not necessarily confirmed by this validation.

Perhaps a summary figure summarising the number of tumours, slides and tiles used in the training and validation cohorts would help assist the reader in understanding the study design.

The authors also used the algorithm and transcriptomic data for MSI prediction. Here the authors do not explain how the ground truth MSI status was determined in the training set. The authors have also not considered how their algorithm compares to standard methods of MSI determination such as immunohistochemistry for MMR proteins or use of PCR. It is also unclear what the purpose of using transcriptomic data in determining MSI status would be. Transcriptomic data is not routinely available in the clinical setting and from a research perspective NGS methods already provide highly accurate methods of determining MSI status, so a clearer explanation of the utility of this application is needed.

-Authors need to explain better how number k of the highest tile prediction is selected.

-Furthermore it would be useful to add a figure that will represent the architecture of the HE2RNA and to justify from which layer the features have been extracted.

-Moreover, how did they specify the organs for which they performed fine-tuning and why didn't they perform fine-tuning for all the cases?

-Finally, in order to prove the potential of the proposed method, it is necessary for authors to add a detailed comparison, providing comparison tables, using state-of-the-art neural networks.

Reviewer #2 (Remarks to the Author):

Schmauch et al developed a deep learning model, HE2RNA, that uses H&E stained whole-slide images (WSIs) to predict gene expression, both as 'bulk' levels and spatially. They also use HE2RNA to predict microsatellite instability (MSI) status in tumours.

Generally, there is a need for these types of models. H&E staining of tumours is routinely performed in the clinic for diagnosis and to guide treatment. Additional tests (e.g.

immunohistochemistry) may be performed for these purposes, but these may be expensive or it may not be possible to perform them at all if there are only limited amounts of tumour tissues available. Predictive machine learning models could help save costs, labour, valuable time and valuable materials.

The authors develop HE2RNA and apply it to H&E stained slides for three purposes:

1. To predict pathway activities and cancer hallmark features in tumours.
2. To perform spatial virtual staining of tumours.
3. To predict MSI status of tumours.

In my opinion, purpose #2 is currently the most valuable aspect of the manuscript. As detailed below, I recommend that the authors provide more details and justifications, in particular with respect to purposes #1 and #3. It is not yet clear to me if/how HE2RNA represents an improvement over existing models in these areas.

Major points

(1) Tumour purity and complexity.

How did the authors handle tumour purity? Aside from tumour cells, tumour tissues may contain: (a) tumour infiltrating lymphocytes and other tumour-associated non-tumour cell types and (b) adjacent normal tissue. (c) Sometimes tumours, or parts thereof, can become necrotic (for example due to low/lack of blood supply). The presence of these three reduce tumour purity and 'dilute' overall 'bulk' gene expression levels. This could influence the prediction of gene expression. (d) Similarly, there may be intra-tumour heterogeneity with clonal and subclonal differences in gene expression.

Could the authors please detail how they handled these four phenomena during model training and validation? For example, was adjacent normal tissue identified and excluded before or after partitioning into tiles? Or was adjacent normal tissue included?

(2) Added value of HE2RNA in prediction of immune cell infiltration and proliferation.

Schmauch et al use gene expression, predicted from H&E stained sections, to determine enrichment in pathway activities and cancer hallmark features. This provides insights in immune cell infiltration, proliferation (cell cycle enrichment) and angiogenesis. It is not yet clear if/how HE2RNA represents an improvement over existing models, such as the frequently used tool VGG16 (Simonyan and Zisserman, 2014, <https://arxiv.org/abs/1409.1556>) or a reportedly better performing tool (Saltz et al 2018 Cell Reports 23:181-193), both for predicting immune cell infiltration. Could the authors compare their AUCs to those from existing models? This would determine the value of HE2RNA in these areas compared to tools that already exist in the field.

(3) Added value of HE2RNA/transcriptomic information in MSI prediction.

Regarding MSI prediction, the authors state in the abstract that "our results show that better performance can be achieved in this setting." A recent paper (Kather et al 2019, Nature Medicine 25:1054-1056) also developed a machine learning model to predict MSI status from WSIs but without involvement of transcriptomic information. I would like to know if gene expression information, a cornerstone of HE2RNA, adds value to MSI status prediction to achieve better performance, as the authors suggest.

To test this, could the authors please compare HE2RNA performance to the performance of the deep learning model developed by Kather et al? Both papers use TCGA data, so it seems feasible to perform a direct comparison. Kather et al report patient-level AUCs of 0.81, 0.84 and 0.77 for the TCGA STAD, CRC-KR and CRC-DX datasets, respectively. How do the AUCs of HE2RNA compare to these? Only a direct comparison can establish whether HE2RNA represents an advance in the field.

(4) Could the authors please justify why they opted to build a deep learning model, instead of

similar models, such as random forest or support vector machines? Is it possible that other models perform better?

Minor points

(5) To facilitate readability of Fig. 1, could the authors add arrowheads to the pink and blue lines to show direction (similar to Fig. S1)?

(6) Could the authors please add scale bars to the images in Fig. 4a,b,d,e, and a colour scale legend to Fig. 4a (similar to Fig. 1, bottom right, but with units added)?

(7) Fig. 4b: There is very little contrast in this image, making it difficult to compare it to the image in panel (a). Can the authors increase the contrast, so that the brown IHC stain is more prominent? Alternatively, could they add inset enlargements of two regions with low/high expression, respectively?

(8) Table S4: How do the expression levels of each of the listed genes contribute to the signatures? Which genes contribute positively, which negative and was weighting applied (if so, please list the weights for each gene) or did all contribute equally?

(9) The authors use CD3 expression as an example to validate their model using IHC. Could the authors add a statement in the discussion to acknowledge that mRNA expression levels do not always correlate with protein levels? It is important to recognise that this is a limitation and that while HE2RNA could be a useful tool, it cannot (yet) necessarily substitute IHC.

(10) In the methods, could the authors please include additional citations for several methods/tools, such as Otsu algorithm, ResNet50, ImageNet dataset? What does ReLu (page 19) stand for?

Reviewer #3 (Remarks to the Author):

The authors present an approach that aims to derive a transcriptomic representation from digital H&E images, which they use for transcriptome prediction, virtual spatialization and to improve prediction of transcriptomic features. They use an approach to glean gene expression from H&E pathology images, using a model (HE2RNA) trained across multiple cancer types. While the authors demonstrate that HE2RNA is capable of identifying cancer type-specific pathways (such as CHK expression in breast cancer), most of their findings highlight the importance of immune components across cancer types (such as prediction of CD3 expression from H&E slides). However, H&Es have previously been used to score immune cells to predict prognosis in multiple cancer types. Therefore it is also not surprising the identified transcriptomic pathway are immune-related pathways. Additionally, many publications have already investigated how lymphocyte scores derived from H&E are correlated with gene expression.

General comment

Given the diverse nature of the journal's readership, could the authors perhaps clearly mention (perhaps in the methods section) how exactly they infer gene expression per tile? This would greatly compliment the otherwise well-explained method section. In this regard: Figure 1: does the cartoon table in the centre (6x6) refer to the training data? If yes, why do Tiles (Tile 1.. N) for ID1 have varying gene expression values? i.e. in the training data, the tile-level gene expression values should be the same for all tiles, for a given sample, is that correct?

Specific comments

Page 4

Paragraph 3: Could the authors clarify if this correlation test/multiple correction between real and

predicted genes done across all cancer types or per cancer type (as depicted in figure 2a)? As a minor point: In figure 2a, can the righthand-Yaxis label be changed to "Number of genes" (since the inset legend already says "#genes with corrected p-values<0.05") for ease of understanding? Paragraph 4: The authors state "The number of significantly well-predicted genes varied considerably between cancer types, mostly due to the size of the dataset considered (Fig. 2a): the smaller the number of samples in the dataset, the higher the correlation coefficient R required for statistical significance."

To substantiate this finding, could the authors show the size of data set for each cancer type? i.e. n=?? for DLBC, MESO etc on the x axis labels. This is available in the methods, but would be helpful to have here

Page 5

Paragraph 1: according to figure 2b, # predicted genes with adj p<0.05 is 0 at 16 cancer types. Why is the x-axis for 16-28 not displayed? This is necessary to substantiate the statement made: ("None of the genes were well-predicted in all 28 available cancer types").

In line 7 of paragraph 1: C1Q (A and B) can be mentioned as CIQA and CIQB to follow conventional gene names

Paragraph 2: any reason these 12 cancer types were chosen? Why not check the overlap between genes predicted across multiple cancer types ref fig 2b? For instance: would the 156 genes predicted in the 12 subtypes be a subset of the ~104 genes predicted in 4 cancer types? Would it be more biologically interpretable to plot the number of unique predicted genes (@adj p<0.05) per cancer type?

Paragraph 3: does this finding relate to the 12 cancer types in which these immune genes had high predicted performance? For example, are any of these 12 cancer types known to have a significant immune component? On the other hand, If any of these 12 cancers are not previously known to be immune-associated, wouldn't this be a novel finding?

Page 6

Paragraph 1&2: The authors have taken a 'candidate' approach by selecting specific cancer signalling pathways and assessing their predicted values from HE2RNA. This analysis is commendable, however:

- The % of cancer types in which a particular pathway was significantly better predicted by HE2RNA is stated (highest being for B and T cell immunity followed by hypoxia, DNA repair, cell cycle and angiogenesis). Could this be a reflection of the number of genes curated for each pathway (refer Supp Table S4- T and B cell signalling have the longest list of constituent genes)? Though the authors have addressed this partially by performing the Rp R0 analysis for a list of random genes, this might not account for the fact that the pathways themselves have variable gene numbers.

Page 7

Paragraph 1: in "HE2RNA, a tool for virtual spatialization" : line 7/8 : how was the 'importance score' used to produce the VSM computed? Is it simply the predicted expression value of the gene(s) of interest (e.g. CD3 genes)?

Paragraph 1, Line 21 : could the authors provide the p value for aggregated real and predicted CD3 expression?

Line 28: does the R value correspond to Figure 4c? if yes, could the authors state R and p val

within scatter plot. Also, axes labels for 4c can be clarified for ease of understanding as: xaxis = predicted CD3 expression ; yaxis=Number of cd3+ T cells from IHC

Reviewer #1 (Remarks to the Author):

1.1 In this manuscript the authors provide an approach to use of machine learning for predicting RNA-seq profiles from histopathology images. There is good clinical rationale for such research to identify predictive/prognostic molecular subtypes from standard H&E stained slides and potentially avoid more expensive molecular tests in certain situations. The strengths of the study are that a large TCGA dataset is used including both training and validation cohorts. A potential limitation is that since majority of the data has been developed from TCGA data there may be inherent artefacts that limit the applicability to other datasets. Ideally validation using a non-TCGA dataset should also be performed.

We thank the reviewer for their insightful comment and reviews of our manuscript. We agree that TCGA has its limitations and we did our best to validate all our findings on additional datasets. In the revised manuscript we have used non-TCGA validation datasets and we have added the results to each corresponding part of the result section in the revised manuscript.

First, we used an additional external cohort, from the study by Kather et al (PMID: 31160815) with per-tile labels for 100,000 tiles for validating the spatialization of lymphocytes. We compared the average predicted expression of the considered genes on tiles labelled with lymphocytes and on tiles from other categories. This prediction allows to distinguish very well tiles containing lymphocytes, as measured by the area under the ROC curve (AUC). The AUC for distinguishing tiles with lymphocytes from other classes is 0.94, while the AUCs computed separately for every class versus lymphocytes range from 0.87 (lymphocytes vs adipose) to 0.98 (lymphocytes vs mucus). This shows that the model, trained only with sample-wise labels, is able to make relevant prediction at the level of individual tiles.

Then, we used an additional double-stained slide HE-CD20 from an LIHC patient from Henri-Mondor University Hospital, using a similar experimental approach than for HE-CD3 double staining. We applied a model predicting *CD3* genes expression and a model predicting *CD19* and *CD20* (specific of B cells) expressions to both slides. This analysis was performed to determine whether these predictive models were specific to different types of lymphocytes or not. For the HE-CD3 slide, the results show that, although both models' predictions correlate positively with the number of T cells, the model trained specifically to predict the expression of *CD3* genes outperforms the model trained to predict *CD19* and *CD20* expressions. Inversely, for the HE-CD20 slide, the model trained to predict *CD19* and *CD20* expressions performs significantly better for detecting B cells than T cells.

See below the table summarizing those results (Table 2), as well as the corresponding panels of revised Figure 4 and Supplementary Figure 6:

Table 2:

	T cell model	B cell model
	0.51	0.39

Correlation coefficient with # T cells on the H&E / CD3 - stained slide			
AUC for distinguishing tiles with # T cells above threshold on the H&E / CD3-stained slide	threshold = 10 T cells (75th percentile)	0.79	0.73
	threshold = 20 T cells (90th percentile)	0.86	0.79
	threshold = 25 T cells (95th percentile)	0.89	0.83
	threshold = 51 T cells (99th percentile)	0.93	0.90
Correlation coefficient with # B cells on the H&E / CD20 - stained slide		0.19	0.23
AUC for distinguishing tiles with # B cells above threshold on the H&E / CD20 - stained slide	threshold = 0 B cells (75th percentile)	0.68	0.66
	threshold = 1 B cells (90th percentile)	0.74	0.76
	threshold = 3 B cells (95th percentile)	0.78	0.82
	threshold = 11 B cells (99th percentile)	0.81	0.89

Figure 4:

Supplementary Figure 6:

We also used an external dataset of 62 slides from prostate adenocarcinoma samples to validate the spatialization of the expression of 3 epithelium-associated genes (*TP63*, *KRT8* and *KRT18*) (PMID: 30696866). Our results demonstrated that the predictions of HE2RNA for these 3 genes per tile correlate significantly with the presence of epithelium, measured by the fraction of pixels belonging to the segmentation mask ($R_{\text{tile}} = 0.40$).

See below the results, added in the Figure 5 of the revised manuscript:

Finally, we applied a model predicting the expression of *MKI67* to an external cohort of 369 liver hepatocarcinoma slides to compare its predictions with pathologist annotations of tumor areas (PMID: 32108950). We found that tiles where the model predicts a significant overexpression of *MKI67* are almost always located in the tumor area: among the 10,000 (resp. 100,000 tiles) over 5 million with the highest predictions, 94% (resp. 90%) are in the tumor. We also found that this overexpression pattern of *MKI67* in the tumoral part of a slides tends to be stronger for patients with an advanced stage tumor. This is consistent with the fact that the *MKI67* labeling index is known to correlate with tumor growth rate (PMID: 7564384) and histological stage (PMID: 7545866).

See below the results, added in the Figure 5 of the revised manuscript:

e.

1.2 It is noted however that validation using CD3 double-stain on a liver hepatocellular carcinoma series is provided, however details on this independent dataset should be more clearly provided i.e. number of tumours, number of slides and how the processing was performed.

This validation approach using staining for the T cell marker CD3 and validating that number of T-cells present in a tile correlates with expression of CD3 RNA seems a good validation step. However, whether the algorithm is equally successful at identifying tumour specific molecular alterations in individual image tiles is not necessarily confirmed by this validation. Perhaps a summary figure summarising the number of tumours, slides and tiles used in the training and validation cohorts would help assist the reader in understanding the study design.

Thank you for this comment, we have clarified this statement in the method section of the revised manuscript. The methods section now reads: *“The single double-stained HE-CD3 slide..”* and in the main text: *“We further validated the spatialization of the T-cell model on a single external H&E-CD3 double stained slide from a LIHC sample (Fig. 4a)”*. We were not able to obtain more double-stained slides as this experimental approach is not done routinely by pathologists. We agree with the reviewer that validation of our models on external cohorts is a major point of approval of our results. We addressed the validation of the models on additional cohorts in the previous point.

1.3 The authors also used the algorithm and transcriptomic data for MSI prediction. Here the authors do not explain how the ground truth MSI status was determined in the training set.

Thank you for your comment, MSI status was determined in the training set using the labels available from the following paper TCGAblinks (PMID: 26704973). In the revised methods we have clarified how the ground truth MSI status was obtained. The manuscript now reads “MSI status data obtained from TCGAblinks (PMID: 26704973). MSI status is assessed by measuring the lengths of a set of mono- and dinucleotide repeats in tumor and matched normal DNA. The MSI-Mono-Dinucleotide Assay used by the TCGA consists of a panel of four mononucleotide repeat loci (polyadenine tracts BAT25, BAT26, BAT40, and transforming growth factor receptor type II) and three dinucleotide repeat loci (CA repeats in D2S123, D5S346, and D17S250) (PMID: 9823339, PMID: 22810696, PMID: 23636398, <https://wiki.nci.nih.gov/display/TCGA/Microsatellite+data>). Tumors are classified as MSI-high (MSI-H) if more than 40% of the markers are altered, MSI-low (MSI-L) if less than 40% of the markers are altered and microsatellite stable (MSS) if no marker is altered (PMID: 16825502).” (Methods Section, line 3 of paragraph **Data for MSI prediction**)

1.4 The authors have also not considered how their algorithm compares to standard methods of MSI determination such as immunohistochemistry for MMR proteins or use of PCR.

Thank you for this comment. For our study, we could not have access to immunohistochemistry (IHC)-stained slides as ground truth for MSI status so we were not able to run such comparison on TCGA and non-TCGA datasets. However, we agree that the specificity and sensitivity of the algorithm cannot compete with Immunohistochemistry (IHC) /PCR performances. Our study suggest that deep-learning approaches could be used as a pre-screening solution before MSI testing by PCR/IHC. Even if those techniques are not expensive, they are currently not routinely performed in cancer with low MSI penetrance (such as Non small cell Lung cancer), however recent publications have demonstrated the impact of MSI status on response to immunotherapy (PMID: 29020592). In such scenario, a pre-selection of eligible patient could be of high interest.

1.5 It is also unclear what the purpose of using transcriptomic data in determining MSI status would be. Transcriptomic data is not routinely be available in the clinical setting and from a research perspective NGS methods already provide highly accurate methods of determining MSI status, so a clearer explanation of the utility of this application is needed.

Thank you for this comment, as discussed above MSI is a prognostic and predictive biomarker used for prediction of patients response to immunotherapy. Its detection can be made by NGS, or PCR/IHC experiments. However, NGS technologies are expensive and are not routinely available worldwide, limited to high-income country and hospitals (PMID: 30234014). Moreover, even though PCR/IHC screening is cheaper than NGS experiments, they are also not routinely performed in hospitals worldwide. One possibility, as already suggested in Kather et al. 2019 (PMID: 31160815), is to use deep learning to infer MSI status from pathology images, and use it as a pre-screening test, since H&E staining of tumours is usually always routinely performed. Our study demonstrates that molecular information gathered at hospitals where transcriptomic molecular analyses are available could be used to improve the MSI status prediction from WSI, not only in these same centers (as it would be obvious), but, through transfer learning, also in other hospitals where

transcriptomic molecular analyses are not available. This is shown in particular in Fig. 6b and 6c. Thus, we think our transfer learning method could help hospitals where molecular profiling are not available. In the main text, we added the following sentences in the main result section of the revised manuscript to make this clearer: “*Although broadly recommended by medical guidelines, systematic screening for MSI condition is performed at high-volume tertiary care centers, but less frequently in low-income hospitals.*” (paragraph **HE2RNA for microsatellite instability status prediction**) and “*Altogether our results demonstrated that learning and transferring a transcriptomic representation learned on a dataset with such available data could help increase the prediction performances of WSI-based models for a clinical diagnosis purpose such as MSI detection in hospitals were transcriptomic and molecular profiling are not done routinely.*” (last sentence of paragraph **HE2RNA for microsatellite instability status prediction**)

1.6 Authors need to explain better how number k of the highest tile prediction is selected.

Thank you for this comment, we modified this paragraph in the revised method section to clarify out statement. The number k is randomly sampled from a predefined list at every training iteration, and for inference we simply average predictions obtained with every value k from this list. We have modified the Methods section in the revised manuscript as follow:

“*The number k is sampled from the list $L = (1, 2, 5, 10, 20, 50, 100)$ for super-tile-preprocessed data ($n_{\text{tiles}} = 100$), and from the list $L = (10, 20, 50, 100, 200, 500, 1000, 2,000, 5,000)$ for full-scale data ($n_{\text{tiles}} = 8,000$). For instance, if $k=1$, the slide-level prediction for a given gene is the largest (super-)tile prediction for this gene, whereas if $k=n_{\text{tiles}}$, the slide-level prediction is the average of all tile predictions. More generally, the slide-level prediction S for a given gene is defined as follows,*

$$S(k) = \sum_{i=1}^k s_i / k$$

where the s_i are the tile-wise predictions for this gene, ordered from the largest (s_1) to the smallest. This stochastic aggregation increases the difficulty of the task and thus reduces overfitting. As such, it acts as a regularization process. During inference, slide-level predictions obtained with every possible value of k are averaged,

$$S = \sum_{k \in L} S(k) / |L|.$$

This is equivalent to calculating a weighted mean of per-tile predictions, with a greater weight for tiles for which the model predicts high levels of expression. The table below summarizes the weights of tile predictions in this weighted average as a function of their rank.”

We have also added a Table (Table S2) to clarify our statement as follow:

Rank of tile prediction	$n_{\text{tiles}} = 100$	$n_{\text{tiles}} = 8,000$
1	0.269	0.0210
1 to 2	0.126	

2 to 5	0.054	
5 to 10	0.026	
10 to 20	0.011	0.0099
20 to 50	0.004	0.0043
50 to 100	0.001	0.0021
100 to 200	-	0.0010
200 to 500	-	0.0004
500 to 1,000	-	0.0002
1,000 to 2,000	-	8×10^{-5}
2,000 to 5,000	-	2×10^{-5}

Table S2: coefficients of tile predictions in the weighted sum defining the slide-level gene expression predicted by the model during inference, as a function of their rank.

1.7 Furthermore it would be useful to add a figure that will represent the architecture of the HE2RNA and to justify from which layer the features have been extracted.

Thank you for this comment, we have added the architecture of our HE2RNA model in the revised manuscript in the revised Supplementary Figure 1, as below:

1.8 Moreover, how did they specify the organs for which they performed fine-tuning and why didn't they perform fine-tuning for all the cases?

Thank you for this comment. The use of the word “finetuning” was indeed misleading, and we modified the revised Methods section accordingly. It should be reminded that the ResNet50 used for feature extraction was never finetuned, instead feature extraction was performed only once using the weights pretrained on ImageNet. For spatialization of the model's prediction, in order to accelerate training when using the full set of 8,000 tiles per slide, we used the following approach:

- Train a network on all TCGA data using 100 supertiles per slide
- Continue training on organs of interest using all available tiles (up to 8000) per slide (which we referred to previously as “finetuning”)

As explained now in the **Training and evaluation** section of the revised Methods, the organs selected for this part are the ones for which we had external data to validate HE2RNA spatialization (i.e. colon, liver and prostate) as well as two organs with the largest datasets available in TCGA (lung and breast), which were used to increase the size of the training set while keeping the training time reasonable (500sec/epoch for training with 8000 tiles / slide, against 7200 sec with all TCGA data).

1.9 Finally, in order to prove the potential of the proposed method, it is necessary for authors to add a detailed comparison, providing comparison tables, using state-of-the-art neural networks.

Thank you for this comment. To the best of our knowledge, this is the first study performing a systematic prediction of transcriptomic data from whole slide images. However, we performed a comparison of HE2RNA with other ML techniques on the example of *CD3* gene expression prediction. In particular, we applied linear regression, SVM and gradient boosted trees (from XGBoost) to ResNet features (results in the table below). Note however that the aim of the article was not to compare different predictive models but to show that HE2RNA can indeed be used to predict transcriptomic profiles, and that this prediction can be used on external datasets to obtain a virtual spatialization for gene expression, or as an intermediate step to better predict MSI status from histology images.

Model	Average correlation per cancer type
HE2RNA	0.36
Linear regression	0.29
SVM	0.34
XGBoost	0.31

We also tried to train all the layers of the pretrained ResNet50 directly on tile images on TCGA-LIHC (in contrast with the remaining of the present work where we just use the frozen ResNet50 as a feature extractor), however this method is very time- and

resource-consuming (3hrs 30min for one training epoch on 4 GPUs against 12 secs/epochs with HE2RNA), and gives worse results.

Model	Correlation on TCGA-LIHC
HE2RNA (using pretrained ResNet50 features)	0.37
ResNet50 finetuned on tile images	0.14

For the prediction of MSI status, we added a comparison of our performance with the previously published work of Kather & al. (Table 3):

a.

	transcriptomic (ours)	WSIs (ours)	WSIs (Kather & al.)
TCGA-CRC-DX	-	0.82	0.77
TCGA-CRC-KR	-	0.83	0.84
TCGA-STAD	-	0.76	0.81

b.

	transcriptomic (ours)	WSIs (ours)	WSIs (adapted from Kather & al.)
TCGA-CRC-DX	0.81	0.71	0.68
TCGA-CRC-KR	0.79	0.72	0.63
TCGA-STAD	0.66	0.63	0.65

Table 3: AUCs for the prediction of MSI status in two extreme regimes: 100% of the data in hospital B (**a**), or 75% of the data in hospital A and 25% in hospital B (**b**).

Reviewer #2 (Remarks to the Author):

Schmauch et al developed a deep learning model, HE2RNA, that uses H&E stained whole-slide images (WSIs) to predict gene expression, both as 'bulk' levels and spatially. They also use HE2RNA to predict microsatellite instability (MSI) status in tumours.

Generally, there is a need for these types of models. H&E staining of tumours is routinely performed in the clinic for diagnosis and to guide treatment. Additional tests (e.g. immunohistochemistry) may be performed for these purposes, but these may be expensive or it may not be possible to perform them at all if there are only limited amounts of tumour tissues available. Predictive machine learning models could help save costs, labour, valuable time and valuable materials.

The authors develop HE2RNA and apply it to H&E stained slides for three purposes:

1. To predict pathway activities and cancer hallmark features in tumours.
2. To perform spatial virtual staining of tumours.
3. To predict MSI status of tumours.

In my opinion, purpose #2 is currently the most valuable aspect of the manuscript. As detailed below, I recommend that the authors provide more details and justifications, in particular with respect to purposes #1 and #3. It is not yet clear to me if/how HE2RNA represents an improvement over existing models in these areas.

We would like to thank the reviewer for their insightful comment and reviews of our manuscript. We think that the three aspects provide different and important value to the manuscript but, as the reviewer suggested, in the revised manuscript, we have emphasized purpose #2 with additional validation cohorts.

Major points

2.1 Tumour purity and complexity.

How did the authors handle tumour purity? Aside from tumour cells, tumour tissues may contain: (a) tumour infiltrating lymphocytes and other tumour-associated non-tumour cell types and (b) adjacent normal tissue. (c) Sometimes tumours, or parts thereof, can become necrotic (for example due to low/lack of blood supply). The presence of these three reduce tumour purity and 'dilute' overall 'bulk' gene expression levels. This could influence the prediction of gene expression. (d) Similarly, there may be intra-tumour heterogeneity with clonal and subclonal differences in gene expression.

Could the authors please detail how they handled these four phenomena during model training and validation? For example, was adjacent normal tissue identified and excluded before or after partitioning into tiles? Or was adjacent normal tissue included?

Thank you for this comment, We agree that tumor purity can be a significant factor when predicting features related to tumor cells only. In our specific setting, we are predicting bulk gene expression profile which is also a combination of different types of cells other than tumour cells. In this case, it makes sense to use the whole slide image to make the prediction as all cells are contributing to the bulk gene expression level.

2.2 Added value of HE2RNA in prediction of immune cell infiltration and proliferation.

Schmauch et al use gene expression, predicted from H&E stained sections, to determine enrichment in pathway activities and cancer hallmark features. This provides insights in immune cell infiltration, proliferation (cell cycle enrichment) and angiogenesis. It is not yet clear if/how HE2RNA represents an improvement over existing models, such as the frequently used tool VGG16 (Simonyan and Zisserman, 2014, <https://arxiv.org/abs/1409.1556>) or a reportedly better performing tool (Saltz et al 2018 Cell Reports 23:181-193), both for predicting immune cell infiltration. Could the authors compare their AUCs to those from existing models? This would determine the value of HE2RNA in these areas compared to tools that already exist in the field.

Thank you for this comment. We tested the fine-tuning of a CNN on tile image approach on TCGA-LIHC dataset only (dataset on 1M tiles), by fine-tuning a ResNet50 directly on tile images, using the slide labels. This approach is much more time and resource-consuming (3hrs 30min for 1 training epoch on 4 GK80 GPUs, for a single TCGA project; as a comparison, HE2RNA runs in 12sec/epochs on a single GPU for all TCGA data with supertile preprocessing), which makes it impractical for our purposes. Furthermore, this approach overfits very quickly, even with data augmentation and regularization. For instance, for the prediction of the four *CD3* genes, this method achieved a correlation $R=0.14$, to be compared with $R=0.37$ obtained with HE2RNA (see response to comment 1.9).

Model	Correlation on TCGA-LIHC
HE2RNA (using pretrained ResNet50 features)	0.37
ResNet50 finetuned on tile images	0.14

2.3 Added value of HE2RNA/transcriptomic information in MSI prediction.

Regarding MSI prediction, the authors state in the abstract that "our results show that better performance can be achieved in this setting." A recent paper (Kather et al 2019, Nature Medicine 25:1054-1056) also developed a machine learning model to predict MSI status from WSIs but without involvement of transcriptomic information. I would like to know if gene expression information, a cornerstone of HE2RNA, adds value to MSI status prediction to achieve better performance, as the authors suggest.

To test this, could the authors please compare HE2RNA performance to the performance of the deep learning model developed by Kather et al? Both papers use TCGA data, so it seems feasible to perform a direct comparison. Kather et al report patient-level AUCs of 0.81, 0.84 and 0.77 for the TCGA STAD, CRC-KR and CRC-DX datasets, respectively. How do the AUCs of HE2RNA compare to these? Only a direct comparison can establish whether HE2RNA represents an advance in the field.

Thank you for this comment, we compared our results to the ones from Kather et al study (PMID: Nature Medicine 25:1054-1056). We demonstrated that at baseline, when the entire colorectal cancer dataset from TCGA is used for predicting MSI (i.e.100% of the data is in hospital B), the AUC obtained by our simple classifier trained directly on WSIs (more precisely on the average of ResNet features over all tiles) is slightly higher than that obtained in the Kather et al. seminal paper on CRC-DX, and comparable on CRC-KR and TCGA-STAD. It shall be noted however that our repeated cross-validation procedure provides robust confidence intervals, taking into account multiple fold-splitting of the data, while Kather et al. results are limited to only one specific split between train and test subsets. Nonetheless, notice that the main aim of this part of our work is to show how the transfer learning of the transcriptomic representation could improve the results of a model predicting MSI, with respect to a classifier learning only and directly from images, when only a small

dataset is available for the task under study. In order to check if this improvement observed was only relative to the simple model under study (an MLP applied to the average of ResNet features), we have also adapted the method of Kather et al. to the setting where only a small amount of data with ground truth MSI status is available at hospital B. As explained in the Supplementary information, we also tried to improve Kather et al. model in this particular setting, by increasing the number of training epochs.

All in all, our results indicate that a simple classifier, provided with an enriched representation obtained by learning gene expression on a different dataset, outperforms also state-of-the-art models for MSI status prediction, when applied to a small dataset (here, 25% of the corresponding TCGA dataset). We have added this information in our revised Figure 6 and Supplementary Figure 7, as shown below:

Figure 6:

Supplementary Figure 7:

2.4 Could the authors please justify why they opted to build a deep learning model, instead of similar models, such as random forest or support vector machines? Is it possible that other models perform better?

Thank you for your comment. We would like to emphasize that such global gene expression predictions are intrinsically multitask efforts, since we are trying to predict simultaneously multiple genes, which motivates the use of a neural network, that can learn to predict several targets at the same time. This property is desirable both for efficiency (no need to train one separate model per gene) and performance (it has been observed that learning from several related task often improves performance on every single task, see e.g.

DOI:10.1023/A:1007379606734). From a computational point of view, it can also be noted that our architecture for learning simultaneously all 30,839 genes, which is a neural network with input dimension 2048 and 3 layers of respective sizes 1024, 1024 and 30,839, has in fact fewer parameters (around 35 million) than what would be required for training one SVM or one linear regression per task (more than 60 million).

We also performed a test on the particular case of the four *CD3* encoding genes. In this scenario, HE2RNA achieved an average correlation per gene and per cancer type $R=0.36$. All other methods we tried performed worse with $R=0.29$ for a linear regression, $R=0.34$ for an SVM and $R=0.31$ for XGBoost.

Finally, it can be noted that the flexibility of neural networks is a major advantage, that allows deconstructing a model trained on whole-slide images to perform spatialization at the level of individual tiles, or to extract a high-level representation from an intermediate layer, as emphasized in paragraph **HE2RNA for microsatellite instability status prediction**.

Minor points

2.5 To facilitate readability of Fig. 1, could the authors add arrowheads to the pink and blue lines to show direction (similar to Fig. S1)?

Thank you for this comment, figure 1 has been modified following your helpful suggestion in the revised manuscript. See the updated figure below.

2.6 Could the authors please add scale bars to the images in Fig. 4a,b,d,e, and a colour scale legend to Fig. 4a (similar to Fig. 1, bottom right, but with units added)?

Thank you for this comment, revised Figure 4 has been modified following your helpful suggestion. See the updated figure below.

2.7 Fig. 4b: There is very little contrast in this image, making it difficult to compare it to the image in panel (a). Can the authors increase the contrast, so that the brown IHC stain is more prominent? Alternatively, could they add inset enlargements of two regions with low/high expression, respectively?

Thank you for this comment, Figure 4b has been modified following your helpful suggestion. See the updated figure below.

2.8 Table S4: How do the expression levels of each of the listed genes contribute to the signatures? Which genes contribute positively, which negative and was weighting applied (if so, please list the weights for each gene) or did all contribute equally?

Thank you for this comment, as we did not have RNAseq from liver of healthy patients we did not know if each gene of each pathway was overexpressed or underexpressed by the analyzed patients. To keep an unsupervised model we then decided to not to apply any weight to any gene from the lists. Since the use of the term “signature” can be misleading, we modified the revised text in the main section, which now reads: *“Based on these hallmarks, we combined several gene signatures from Gene Set Enrichment Analysis (GSEA) software for each of these biological networks, to obtain six lists of genes involved in pathways known to be deregulated in several types of cancers: increased angiogenesis, increased hypoxia, deregulation of the DNA repair system, increased cell cycle activity, immune response mediated by B cells and adaptive immune response mediated by T cells (see Methods and Table S7).”*

2.9 The authors use CD3 expression as an example to validate their model using IHC. Could the authors add a statement in the discussion to acknowledge that mRNA expression levels do not always correlate with protein levels? It is important to recognise that this is a limitation and that while HE2RNA could be a useful tool, it cannot (yet) necessarily substitute IHC.

Thank you for this comment, we agree this is an important point. We further agree that the next step would be to validate our HE2RNA model on proteomic data and determine what is the actual correlation between transcriptomic expression and protein expression in such context. We have added this statement and the revised manuscript now reads: *“However, recent studies (PMID: 24082820) have shown that mRNA and protein expressions may be poorly correlated due to various factors such as different half lives and post-transcription machinery. Thus, a joint analysis of the transcriptomic and proteomic data could provide useful insights and increase the overall performance of our model.”*

2.10 In the methods, could the authors please include additional citations for several methods/tools, such as Otsu algorithm, ResNet50, ImageNet dataset? What does ReLu (page 19) stand for?

Thank you for this comment, ReLU stands for Rectified Linear Unit and follow this formula: $\text{ReLU}(x) = 0$ if $x < 0$ and $\text{ReLU}(x) = x$ if $x > 0$ (arXiv:1611.01491). We have added the definition and additional citations for all the tools used in our study in the revised Methods section. ResNet50 (10.1109/CVPR.2016.90), ImageNet (DOI:10.1109/CVPR.2009.5206848), Otsu algorithm (DOI: 10.1109/TSMC.1979.4310076).

Reviewer #3 (Remarks to the Author):

The authors present an approach that aims to derive a transcriptomic representation from digital H&E images, which they use for transcriptome prediction, virtual spatialization and to improve prediction of transcriptomic features. They use an approach to glean gene expression from H&E pathology images, using a model (HE2RNA) trained across multiple cancer types. While the authors demonstrate that HE2RNA is capable of identifying cancer type-specific pathways (such as CHK expression in breast cancer), most of their findings highlight the importance of immune components across cancer types (such as prediction of CD3 expression from H&E slides). However, H&Es have previously been used to score immune cells to predict prognosis in multiple cancer types. Therefore it is also not surprising the identified transcriptomic pathway are immune-related pathways. Additionally, many publications have already investigated how lymphocyte scores derived from H&E are correlated with gene expression.

We thank the reviewer for their detailed review and appreciate their insightful comments on the paper. Our main aim is to demonstrate that machine learning models can learn new predictive features from histology images only without localized region annotations from medical experts. We agree there are already some studies published in peer reviewed journal about prediction of immune score using similar approaches. This is why we also demonstrated that HE2RNA and the transcriptomic learning could be used for other clinical questions such as predicting MSI status from biopsy images, and that it could be particularly useful when only few labeled examples are available for the learning part. In the revised manuscript we have also demonstrated that HE2RNA could be used to predict other histological pattern beside immune infiltration, such as, for example, epithelial cells, which are important to define tumoral stroma.

3.1 General comment

Given the diverse nature of the journal's readership, could the authors perhaps clearly mention (perhaps in the methods section) how exactly they infer gene expression per tile? This would greatly compliment the otherwise well-explained method section.

Thank you for this comment. We clarified this in the method section, which now reads:
"As mentioned above, the model produces one score per tile and per gene. For each gene, the predictions per tile are subsequently aggregated to produce one prediction per gene at the level of the slide. To generate Virtual Spatialization Maps, we simply omit this aggregation step and interpret the score of a tile as the predicted gene expression for this tile."

3.2 In this regard: Figure 1: does the cartoon table in the centre (6x6) refer to the training data? If yes, why do Tiles (Tile 1.. N) for ID1 have varying gene expression values? i.e. in the training data, the tile-level gene expression values should be the same for all tiles, for a given sample, is that correct?

Thank you for this comment, the cartoon table refer to the predicted values for each tile for each gene for each patient that is being used for the transcriptomic learning. The final prediction of the model for the whole slide is then computed starting from these local predictions and can be compared with the ground truth gene expression prediction for the slide. Nonetheless, this per-tile prediction can be used to provide a prediction for the spatialization of each gene expression, which can be tested for example in double stained images. We clarified it in the Fig.1 legend. which now reads: "*For each predicted coding or non-coding gene a score is calculated for each tile on the corresponding WSI, which can be interpreted as the predicted gene expression for this tile (even though the real value is available only for the slide)*".

3.3 Specific comments

Page 4

Paragraph 3: Could the authors clarify if this correlation test/multiple correction between real and predicted genes done across all cancer types or per cancer type (as depicted in figure 2a)? As a minor point: In figure 2a, can the righthand-Yaxis label be changed to "Number of genes" (since the inset legend already says "#genes with corrected p-values<0.05") for ease of understanding?

Paragraph 4: The authors state "The number of significantly well-predicted genes varied considerably between cancer types, mostly due to the size of the dataset considered (Fig. 2a): the smaller the number of samples in the dataset, the higher the correlation coefficient R required for statistical significance."

To substantiate this finding, could the authors show the size of data set for each cancer type? i.e. n=?? for DLBC, MESO etc on the x axis labels. This is available in the methods, but would be helpful to have here

Thank you for this comment, the correlation between real and predicted genes is corrected per cancer type, taking into account the fact that we are predicting the expressions of 30,839 genes. We have clarified this statement in the result section of the revised manuscript.

The methods section now reads: "*This correction was performed separately for each cancer type.*"

We have also modified the Figure 2a in the revised manuscript as shown below:

3.4 Page 5

Paragraph 1: according to figure 2b, # predicted genes with adj p<0.05 is 0 at 16 cancer types. Why is the x-axis for 16-28 not displayed? This is necessary to substantiate the statement made: (“None of the genes were well-predicted in all 28 available cancer types”). In line 7 of paragraph 1: C1Q (A and B) can be mentioned as CIQA and CIQB to follow conventional gene names

Thank you for this comment, in the revised manuscript we have modified the figure 2a and supplementary figure S2 see below:

We have also clarified the text concerning the C1Q genes nomenclature. The revised manuscript now reads: “*CQ1A and CQ1B proteins*”.

3.5 Paragraph 2: any reason these 12 cancer types were chosen? Why not check the overlap between genes predicted across multiple cancer types ref fig 2b? For instance: would the 156 genes predicted in the 12 subtypes be a subset of the ~104 genes predicted in 4 cancer types? Would it be more biologically interpretable to plot the number of unique predicted genes (@adj p<0.05) per cancer type?

Thank you for this comment, we did not choose any cancer, we looked at the genes that were significantly well predicted in at least 12 cancers, which are not necessarily the same 12 cancers for each of those genes. To clarify this statement, the revised manuscript now reads: “*We found 156 genes that were well predicted separately in at least 12 out of 28 different cancer types. For this subset of genes, we performed a functional annotation (Fig. 2c)*” (line 3 of this paragraph). The 156 genes well predicted on > 12 cancers are a subset of the 10,000 well predicted on > 4 cancers.

Following your suggestion, we analyzed unique well predicted genes per cancer type (plot below). However, the analysis of those genes with Gene Set Enrichment Analysis (GSEA) software and Molecular Signatures Database v6.2 did not reveal any interesting pattern. Indeed, for most cancer types, unique well-predicted genes (that include many non-coding fragments) do not overlap significantly with any known gene signatures. An exception is liver hepatocarcinoma (TCGA-LIHC), where 82 unique genes are significantly well predicted,

which include 12 genes from the “Hsiao liver specific genes” (liver selective genes, PMID: 11773596, p-value = 7.27×10^{-15}) and 11 genes from the “Chiang liver subclass proliferation down” (Top 200 marker genes down-regulated in the 'proliferation' subclass of hepatocellular carcinoma (HCC); characterized by increased proliferation, high levels of serum AFP, and chromosomal instability, PMID: 30413412, p-value = 7.54×10^{-15})

3.6 Paragraph 3: does this finding relate to the 12 cancer types in which these immune genes had high predicted performance? For example, are any of these 12 cancer types known to have a significant immune component? On the other hand, If any of these 12 cancers are not previously known to be immune-associated, wouldn't this be a novel finding?

Thank you for this comment, as we mentioned in our previous comment, the 12 cancers are not fixed, instead we selected genes well-predicted on at least 12 different cancer types, so the list is different for each gene. However, it can be noted that among those, 10 cancers are more frequently encountered:

- TCGA-BLCA: 153 well predicted genes from the 156 (for a total of 5,557 well predicted genes for this cancer)
- TCGA-BRCA: 155 (10,100)
- TCGA-CESC: 133 (4,426)
- TCGA-HNSC: 136 (8,030)
- TCGA-KIDN: 140 (7,542)
- TCGA-LIHC: 149 (3,768)
- TCGA-LUNG: 156 (15,391)
- TCGA-STAD: 127 (3,402)
- TCGA-THCA: 141 (5,912)
- TCGA-UTER: 139 (8,706)

Unsurprisingly, those are the cancers with the largest number of significantly well predicted genes. Moreover, all these tumors were described among tumour with high leukocyte infiltration by Throsson et al (PMID: 29628290). Especially TCGA-LUNG, TCGA-HNSC, TCGA-STAD, TCGA-CESC are among the top ten. Recent studies also demonstrated that

DNA repair defects in cancer cells were associated with the host immune response and tumor susceptibility to immunotherapy (PMID: 30272580). Such DNA repair defect were found in breast, endometrial, cervical and colorectal cancer. Altogether it seems that among the 10 cancers are more frequently encountered all have already been demonstrated to be influence by the immune infiltration.

3.7 Page 6

Paragraph 1&2: The authors have taken a 'candidate' approach by selecting specific cancer signalling pathways and assessing their predicted values from HE2RNA. This analysis is commendable, however:

- The % of cancer types in which a particular pathway was significantly better predicted by HE2RNA is stated (highest being for B and T cell immunity followed by hypoxia, DNA repair, cell cycle an angiogenesis). Could this be a reflection of the number of genes curated for each pathway (refer Supp Table S4- T and B cell signalling have the longest list of constituent genes)? Though the authors have addressed this partially by performing the Rp R0 analysis for a list of random genes, this might not account for the fact that the pathways themselves have variable gene numbers.

Thank you for this comment, we agree this could slightly influence the performance, this is why we compared the predictions obtained for the genes in the signatures with those obtained for 10,000 different random lists of genes of the same length as the signature under study. Moreover, the number of genes varies between 28 and 78 genes, which is a small variation compared to the total number of coding genes that are predicted and explain why we did not take it into account.

3.8 Page 7

Paragraph 1: in "HE2RNA, a tool for virtual spatialization" : line 7/8 : how was the 'importance score' used to produce the VSM computed? Is it simply the predicted expression value of the gene(s) of interest (e.g. CD3 genes)?

Thank you for this comment, for simplicity we used the predicted value of the gene of interest. As explained in more details in the revised method section (named **Per gene inference**) of the manuscript, the inference of gene expression per tile is straightforward. The model, by design, generates one score per tile and per gene, then aggregates those scores through a weighted average to generate one prediction per gene at the level of the slide, so we just extracted the scores per tile and interpreted them as the local prediction for gene expression.

3.9 Paragraph 1, Line 21 : could the authors provide the p value for aggregated real and predicted CD3 expression?

Thank you for this comment, we added a table with detailed results and p-values for the genes used for virtual spatialization. See below:

gene	correlation (R)	p-value (HS)	p-value (BH)
TCGA-LIHC			
CD3D	0.40	$< 10^{-4}$	$< 10^{-4}$
CD3E	0.41	$< 10^{-4}$	$< 10^{-4}$
CD3G	0.41	$< 10^{-4}$	$< 10^{-4}$
CD247	0.37	$< 10^{-4}$	$< 10^{-4}$
CD19	0.32	$< 10^{-4}$	$< 10^{-4}$
CD20	0.27	0.6	2×10^{-4}
MKI67	0.47	$< 10^{-4}$	$< 10^{-4}$
TCGA-COAD			
CD3D	0.43	0.6	8×10^{-4}
CD3E	0.39	0.6	8×10^{-4}
CD3G	0.41	0.6	8×10^{-4}
CD247	0.39	0.6	8×10^{-4}
CD19	0.20	~1	0.009
CD20	0.11	~1	0.07
TCGA-PRAD			
TP63	0.18	~1	0.009
KRT8	0.12	~1	0.06
KRT18	0.12	~1	0.07

3.10 Line 28: does the R value correspond to Figure 4c? if yes, could the authors state R and p val within scatter plot. Also, axes labels for 4c can be clarified for ease of understanding as: xaxis = predicted CD3 expression ; yaxis=Number of cd3+ T cells from IHC

Thank you for this comment, we have clarified the corresponding panel in the revised Figure 4 of the manuscript. See below:

REVIEWER COMMENTS

Reviewer #1 (Remarks to the Author):

I have reviewed the rebuttal alongside the revised manuscript by Schmauch and coworkers. Overall I'm impressed with the additional validation work which had been suggested by myself as well as reviewers #2 and #3. These studies internally validate the method, although they do not drive discovery (for example the Ki67 labelling in tumoural areas is confirmatory, novel markers are not derived from this).

There are several minor points to address which I feel should improve the final manuscript

Figure 2a predictably shows that the number of genes which reveal statistically significant correlation of multiple testing correction increases with sample size. One notable outlier with limited predictive power despite large case numbers is COAD (on a par with DLBCL), which is relevant as these are used later as validation for the MSI technique. Can the authors introduce a separate panel wherein the predictive power within the COAD (and STAD?) cohorts is examined after filtering for MSI+ cases, ie. examining only MSS cases? Does the number of correctly predicted genes increase? If so, which genesets? This would in my opinion further indicate this technique can reveal subtype-specific biology rather than correlate to known morphologic features such as lymphocyte infiltration. This could be relevant in follow-up for example correlating immune-evasion gene expression signatures to tumour morphology.

On a tangent to this point, the authors often state correlation coefficients ('CD3 receptor is encoded by four genes: CD3D , CD3E , CD3G and CD247, as these genes were well predicted by HE2RNA model (R COAD = 0.40, R LIHC = 0.40)') with qualifiers ('well predicted') but without accompanying p-value. This is relevant as a rho of 0.4 isnt particularly strong and 'well predicted' is therefore debatable. I trust that in many instances given the large dataset this p-value will be highly significant but it must be stated. Were these T cell receptor subunits part of the (small) geneset correctly predicted for COAD after multiple testing correction?

The abstract states 'as validated by double-staining on an independent dataset'. This should read as 'as validated by CD3 and CD20 labelling on an independent dataset' to clearly outline the limits of the validation exercise.

The paper discusses MKI67 expression. For the non-aficionados it should be made clear that 'clinically this is detected using Ki67 or MIB1 immunohistochemistry' or similar.

At some point (nb. there are no page or line numbers in this submission) the manuscript reads 'and also occur in other cancers (e.g breast, prostate...)' Clearly the dots should be removed here.

At several junctures in the manuscript (see Table 4) the authors describe their data as 'ours'. This is rather blunt. 'This manuscript' would be more elegant.

Likewise the authors refer throughout (rightly so) to the work by Kather and colleagues. In tables this is labelled as 'Kather & al'. The authors should simply state the reference number.

Marnix Jansen, UCL London

Reviewer #3 (Remarks to the Author):

The authors have addressed comments sati. The additional analyses and rewording of the manuscript have made it more lucid.

REVIEWER COMMENTS

Reviewer #1 (Remarks to the Author):

I have reviewed the rebuttal alongside the revised manuscript by Schmauch and coworkers. Overall I'm impressed with the additional validation work which had been suggested by myself as well as reviewers #2 and #3. These studies internally validate the method, although they do not drive discovery (for example the Ki67 labelling in tumoural areas is confirmatory, novel markers are not derived from this).

There are several minor points to address which I feel should improve the final manuscript

Figure 2a predictably shows that the number of genes which reveal statistically significant correlation of multiple testing correction increases with sample size. One notable outlier with limited predictive power despite large case numbers is COAD (on a par with DLBCL), which is relevant as these are used later as validation for the MSI technique. Can the authors introduce a separate panel wherein the predictive power within the COAD (and STAD?) cohorts is examined after filtering for MSI+ cases, ie. examining only MSS cases? Does the number of correctly predicted genes increase? If so, which genesets? This would in my opinion further indicate this technique can reveal subtype-specific biology rather than correlate to known morphologic features such as lymphocyte infiltration. This could be relevant in follow-up for example correlating immune-evasion gene expression signatures to tumour morphology.

We thank the reviewer for their insightful comment. First, we should mention here that we merged READ and COAD into CRC in paragraph **A deep learning model for the prediction of gene expression**, in order to be more consistent with paragraph **HE2RNA for microsatellite instability status prediction**. We checked that this does not affect the conclusions, and modified the numbers wherever necessary. See the updated figure 2 below:

For CRC, when restricting the analysis to MSS patients only, the number of well predicted genes slightly increased (1030 well predicted genes, compared to the 928 well predicted

genes on the whole CRC dataset). It should also be noted that, when restricting the analysis to MSI-H patients only (85 samples), the number of well predicted genes is even higher (1288 genes), much above what could be expected given the size of this subset (in particular if we consider other datasets with similar size, such as MESO or SKCM). As shown in the figure below:

We then performed a gene-sets enrichment analysis of the well predicted genes in MSS-centered patients (see figure below, panel a). In these patients we identified mostly pathways involved in RNA metabolism and Translation regulation (Formation of free 40S subunits, Translation initiation...). Such findings could be of interest to define novel subtypes of CRC cancers. However, more biological validation would be needed and none of these pathways are being part of the four robust consensus molecular subtypes used for CRC classification (MSI immune, Canonical, Metabolic and Mesenchymal) (PMID: 26457759).

A

Interestingly, using a similar analysis in MSI-H-centered patients (see figure below, panel b), we observed in these patients an enrichment of well predicted genes involved in T cell activation and immune activation (PD-1 signaling, Interferon gamma signaling...). These results are confirming the high capacities of HE2RNA to predict immune infiltrate and are

aligned with the known higher immune infiltration observed in these patients and linked to their positive response to immunotherapies.

B

On the STAD dataset, strikingly, we observed the opposite phenomenon, i.e. the number of well predicted genes seems to depend only on the size of the considered subset: 3437 well predicted genes on the whole dataset (371 samples), 1402 genes on MSS (251 samples) and 49 well predicted genes only on MSI-H dataset (61 samples).

Overall, our results seemed consistent with the better results obtained on CRC than on STAD for predicting MSI status in the transfer learning setup. In a further study, it would be interesting to see the performances of CNN models in predicting the consensus molecular subtypes used for CRC classification and run similar gene set enrichment analysis to observe if we can better predict genes involved in those pathways.

As these analysis did not gave us a full answer on the interpretability of HE2RNA performances, we decided not to include it in the present version of manuscript. However if the reviewer thinks it bring additional value to the manuscript we can add it into a supplementary figure.

On a tangent to this point, the authors often state correlation coefficients ('CD3 receptor is encoded by four genes: CD3D , CD3E , CD3G and CD247, as these genes were well predicted by HE2RNA model (R COAD = 0.40, R LIHC = 0.40)') with qualifiers ('well predicted') but without accompanying p-value. This is relevant as a rho of 0.4 isnt particularly strong and 'well predicted' is therefore debatable. I trust that in many instances given the large dataset this p-value will be highly significant but it must be stated. Were these T cell receptor subunits part of the (small) geneset correctly predicted for COAD after multiple testing correction?

We thank the reviewer for their comment. On CRC, those genes (CD3D, CD3E, CD3G and CD247) indeed do not meet the significance requirement under Holm-Sidak correction (as stated earlier, the significance threshold for this dataset is significantly higher than for

datasets of similar size) and our statement was incorrect. The mention “well predicted” referred to the prediction on TCGA-LIHC (as in a previous version we only considered LIHC samples for spatialization), as shown in Table 1.

However, it is worth mentioning that for both cancer types, those genes do meet significance requirement under the less stringent Benjamini-Hochberg correction (Table 1), meaning that the rate of false discoveries is actually expected to be low. Nonetheless, for consistency with the choice we made along the manuscript, we have removed the mention “well predicted”.

The revised Main text (p.6, second paragraph) now reads:

CD3 receptor is encoded by four genes: CD3D, CD3E, CD3G and CD247^{45,46}. We used their prediction (correlations and p-values in Table 1) to define the spatial localisation of the T cells (later named T-cell model). Similarly, to define the B cell population, we considered CD19 and CD20 proteins expressed exclusively by B lymphocytes^{47,48}, and used their prediction (Table 1) to define the spatial localisation of the B cells (later named B-cell model).

Furthermore, all correlations and p-values for genes mentioned in paragraph **HE2RNA, a tool for virtual spatialization** are indicated in table 1.

The abstract states ‘as validated by double-staining on an independent dataset’. This should read as ‘as validated by CD3 and CD20 labelling on an independent dataset’ to clearly outline the limits of the validation exercise.

We thank the reviewer for their comment. We modified this sentence in the revised abstract accordingly.

The paper discusses MKI67 expression. For the non-aficionados it should be made clear that ‘clinically this is detected using Ki67 or MIB1 immunohistochemistry’ or similar.

We thank the reviewer for their comment, we modified the revised text accordingly.

At some point (nb. there are no page or line numbers in this submission) the manuscript reads ‘and also occur in other cancers (e.g breast, prostate...)’ Clearly the dots should be removed here.

We have added the page number and the dots were removed.

At several junctures in the manuscript (see Table 4) the authors describe their data as ‘ours’. This is rather blunt. ‘This manuscript’ would be more elegant.

We thank the reviewer for their comment, we modified the revised text accordingly.

Likewise the authors refer throughout (rightly so) to the work by Kather and colleagues. In tables this is labelled as ‘Kather & al’. The authors should simply state the reference number.

We have modified table 4 accordingly.

Marnix Jansen, UCL London

Reviewer #3 (Remarks to the Author):

The authors have addressed comments satisfactorily. The additional analyses and rewording of the manuscript have made it more lucid.

REVIEWERS' COMMENTS:

Reviewer #1 (Remarks to the Author):

The authors have satisfactorily addressed my queries. I would be very much in favour of the authors NOT lumping COAD (colorectal adenocarcinoma) and READ (rectal adenocarcinoma) under the CRC rubric and, in fact, include in the main text and as a figure panel in the main text the differential analysis of their tool in MSI+ v MSS colorectal adenocarcinoma. Recovery of inflammatory targets in the former is a neat validation of the tool within a tumour category rather than between tumour categories which is where most end users will use their technology.

Marnix Jansen
University College London

Response to reviewer

Reviewer #1 (Remarks to the Author):

The authors have satisfactorily addressed my queries. I would be very much in favour of the authors NOT lumping COAD (colorectal adenocarcinoma) and READ (rectal adenocarcinoma) under the CRC rubric and, in fact, include in the main text and as a figure panel in the main text the differential analysis of their tool in MSI+ v MSS colorectal adenocarcinoma. Recovery of inflammatory targets in the former is a neat validation of the tool within a tumour category rather than between tumour categories which is where most end users will use their technology.

We would like to thank the reviewer for their suggestions. We have updated the manuscript accordingly and did not combine colorectal and rectal carcinomas under a global CRC cohort. We have also included our analysis performed on MSI-H only cohort versus MSS only from patients with colorectal adenocarcinoma in section **HE2RNA for microsatellite instability status prediction:**

The analysis of gene expression prediction, restricted to MSI-H patients from TCGA-COAD (81 samples), revealed that a surprisingly high number of genes were significantly well predicted by HE2RNA on this subset (1027 genes well predicted under HS correction), more than on the whole dataset (324 well predicted genes for 463 samples) or on the subset of MSS patients (592 well predicted genes for 277 samples) (Fig. 6a). A gene set enrichment analysis of the genes well predicted in MSI-H patients revealed an enrichment in T cell activation and immune activation (PD-1 signaling, Interferon gamma signaling...). These results confirmed the high performance of HE2RNA to predict immune infiltrate and are aligned with the known higher immune infiltration observed in these patients and linked to their positive response to immunotherapies (Fig. 6b). Performing a similar analysis in MSS patients, we identified mostly pathways involved in RNA metabolism and Translation regulation (Formation of free 40S subunits, Translation initiation...) (Supplementary Fig. 7).